# Railway Lines across the Alps: Analysis of Their Usage through a New Railway Link Cost Function

**Marino Lupi \*, Antonio Pratelli, Daniele Conte and Alessandro Farina**

University Centre of Logistic Systems, Department of Civil and Industrial Engineering, University of Pisa, 56126 Pisa, Italy; antonio.pratelli@unipi.it (A.P.); dc1996@hotmail.it (D.C.); alessandro.farina@unipi.it (A.F.)
\* Correspondence: marino.lupi@unipi.it

**Abstract:** In this paper, the usage of railway lines across the Alps is evaluated, both at present and after the new lines and base tunnels will be in operation. The railway network of a large part of Europe has been modelled through a graph, and the best routes between some of the most important origin/destination pairs in Italy and Europe have been determined. A new cost function has been developed for the links of the network. The proposed cost function is an improvement of those existing in the literature, because all cost components are taken into account in detail, while the traction cost and the number of locomotives utilized explicitly depend on the geometrical characteristics of rail lines. This last aspect is crucial in analyzing the rail lines across the Alps, as they are often operated in double or triple traction. The results of the study show the importance of new Alpine rail lines and base tunnels: the Ceneri base tunnel will remove a bottleneck on the Gotthard line, while the Brenner and Frejus base tunnels will take up a quota of demand currently served by other lines. Moreover, the new Alpine lines will create an east–west rail connection, through the Italian Padan Plain, alternative to the rail route which currently bypasses the Alps to the north.

**Keywords:** railway freight transport; railway cost function; Alpine passes; railway lines across the Alps; base tunnels

## 1. Introduction

Most railway lines crossing the Alps were built in the nineteen century and are often steep and tortuous. Almost all these lines are operated in double traction. Some of them are operated in triple traction, for example: the Brenner line on the Austrian side, and the Frejus line on both the Italian and the French side. This greatly increases not only monetary costs, but also travel times. On the other hand, several improvement projects have been developed: two new base tunnels have already been opened (the Gotthard and the Lötschberg ones), but several other base tunnels and new lines are currently under construction or planned.

In this paper, a new cost function for railway links has been considered, then, after a deep analysis of the geometry and operation of each line crossing the Alps, the importance of each Alpine railway pass is evaluated.

The minimum cost paths between the most important rail terminals of Italy, from one side, and the most important rail terminals of Central-Southern Europe and of Central-Eastern Europe, from the other side, have been calculated. Which Alpine pass is crossed by each minimum cost path has been determined. Minimum cost routes have been calculated by means of the Dijkstra algorithm. Two scenarios of study have been taken into account, namely: a 'current scenario', which involves only railway lines currently in operation, and a 'project scenario', which involves also railway lines under construction or planned. In order to do so, the railway network of a large part of Europe has been modelled. It comprises northern Italy, large part of France, Germany, Belgium, the Netherlands,

Luxembourg, Austria, Switzerland, Czech Republic, western parts of Slovakia and Hungary, Slovenia, northern Croatia, and northern Serbia.

The analysis of the usage of Alpine passes, presently and after the opening of the new base tunnels and lines, currently under construction and planned, has been composed of the following main phases:

1.  Development of a new monetary cost function for freight railway links;
2.  Analysis of the geometry and operation characteristics of each line crossing the Alps;
3.  Construction of the railway network model of a large part of Europe;
4.  Calculation of the minimum cost paths, based on travel times, as well as monetary and generalized costs, between pairs of Italian and European railway terminals, which require crossing the Alps.

## 2. The Proposed Cost Function

### 2.1. Monetary Cost

In the evaluation of the cost function of a railway network link, two points of view, for the cost, can be considered:

-   cost for the production of the service,
-   cost for purchasing the service.

The cost for the production of the service is, for example, the cost supported by the railway companies to put the train in operation. The cost for purchasing the service is the price that railway companies offer to shippers and customers. In this research, we are interested in the evaluation of the cost for the production of the service.

In this paper, the cost to transport intermodal transport units (ITUs) (containers, swap bodies, semitrailers) has been considered.

#### 2.1.1. State of the Art on Cost Functions of Rail Links

In the literature, several cost functions, which express the railway cost in an aggregate way, can be found (Kim and Van Wee [1], Brummersted et al. [2], Sawadogo et al. [3] and Janic [4]). Less frequent are cost functions which express the railway cost in a detailed way, considering separately: staff cost; amortization, maintenance and insurance costs of locomotives and wagons; cost for the usage of rail track; traction cost (Grosso [5] and Baumgartner [6]).

The advantage of the cost function proposed by Grosso is the higher level of detail than cost functions proposed in [1–4]; but:

-   it does not take into account the different energy consumption for different values of slope, and in particular it does not take into account explicitly the resistances to motion;
-   cost values of the cost function components are not in line with those proposed in the literature, and in particular with those proposed in the Baumgartner cost function, which will be described in the following section;
-   the number of locomotives considered for each train, for each network link, is not clearly stated.

However, the cost function proposed by Grosso has been a departure point for the research presented in this paper.

The cost function proposed by Baumgartner is described in detail in [6]. This cost function takes into account the following cost components:

-   electric locomotive purchase, amortization and maintenance costs [€/(km·locomotive)].
-   flat wagon (for containers) purchase and maintenance costs [€/(km·wagon)].
-   electric traction power consumption costs [€/km].

The cost function proposed by Baumgartner provides details in several cost components, and the proposed cost values are in line with those in the literature: for example, the purchase cost of a locomotive, or the total km travelled in a year by a locomotive or by wagons, are in line with the values commonly considered by rail transport companies and Multimodal Transport Operators (MTOs). On the other hand, Baumgartner's cost function misses some components: the rail track cost, the staff cost, and the locomotives and wagons insurance.

A comparison of the proposed cost function with those by Grosso and Baumgartner is exposed in Table 1.

Another detailed cost function is that proposed in Dolinayovà et al. [7], which has been developed for rail lines in Slovakia. However, it does not take into account explicitly the geometrical characteristics of rail lines in the calculation of the traction cost and of the number of locomotives necessary to operate the train. Moreover, it does not report reference values for all cost components taken into account, but only for some of them, namely the costs related to the wagons.

### 2.1.2. The Proposed Monetary Cost Function

In this paper, monetary costs of railway transport have been calculated basing partially on the research performed by Baumgartner [6], while some reference costs, regarding staff, locomotives and wagons, have been determined basing on [8]. It must be underlined that: the authors determined all resistances to motion in detail, on the analyzed network, which consists of a large part of Europe. The authors considered the different railway track usage prices and the different electricity prices charged by each European country.

The proposed cost function is the following:

$$C\ [€] = t\ [\text{h}]\cdot(n_d\cdot P) + l\ [\text{km}]\cdot\{n_L\cdot(A_L + M_L + I_L) + n_W\cdot(A_W + M_W + I_W) + R + T(V_A, i, R_c)\} + 2\cdot H\cdot n_{ITU} \quad (1)$$

where:

- $C$ = in € per train service, is the monetary cost on each rail link, having length $l$ and travel time $t$;
- $P$ = staff cost [€/(h·driver)]: cost of the train drivers. The staff cost is not the same in the whole Europe: in Italy an average cost per hour, for each train driver, of 35 € was detected (this cost comprises not only the net salary but also pension contributions and healthcare) (Source: Trenitalia, relazione annuale [9]), while in Germany, it resulted in a cost of 42 € per hour per driver [8]. Therefore, for the whole Europe an average staff cost of 38.5 €/h per European driver has been considered.
- $n_d$ is the number of drivers of each freight train (independently of the number of locomotives). Two train drivers per freight train are necessary in Italy, while only one driver is sufficient in the rest of Europe (sources: interviews to Rete Ferroviaria Italiana and [10]);
- $A_L$ = amortization cost of one locomotive. In [8] it is reported that a reference amortization cost for a locomotive used for freight transport is 330,670 € per year. Mercitalia Rail (the main Italian rail freight company) has provided a reference value for the number of km travelled each year by a locomotive: 200,000 km. Therefore, the average amortization cost of a locomotive, expressed in €/(locomotive·km), has been estimated: 1.653 €/(locomotive·km);
- $M_L$ = maintenance cost of one locomotive: in [8] it is suggested to take it as 5.5% of the amortization cost, that is: 0.091 €/(locomotive·km);
- $I_L$ = insurance cost of one locomotive. In [8], it is suggested to take it as 1.5% of the amortization cost: that is, 0.025 €/(locomotive·km);
- $n_L$ = number of locomotives. The number of locomotives depends on the gradient of the railway link, and it ranges from 1 to 3. The calculation of the number of locomotives is described in detail in the following;
- $A_W$ = amortization cost of one wagon; Sgns flat wagons for containers have been considered. In [8] it is reported that the amortization cost for a Sgns is 4898 €/year. Mercitalia Rail has provided

a reference value for the number of km travelled each year by a wagon of 50,000 km. Therefore, the average amortization cost of a wagon, is 0.098 €/(wagon·km);

- $M_W$ = maintenance cost of one wagon: in [8] it is suggested to take it as 10% of the amortization cost: that is, 0.0098 €/(wagon·km);
- $I_W$ = insurance cost of one wagon: in [8] it is suggested to take it as 1.3% of the amortization cost: that is, 0.0013 €/(wagon·km);
- $R$ = railway track cost [€/km], i.e., cost for the usage of the railway infrastructure, paid by the railway transport company to the infrastructure manager. This cost has been determined, for all countries involved in this research, according to the values provided in [11] for Italy and in [12] for the other European countries. The rail track cost is different from one country to another, from a line to another, and it depends also on the weight of the train;
- $H$ = cost of handling at rail terminals [€/ITU]. It is available on the Terminali Italia website (source: Terminali Italia [13]), and it is equal to 32.5 € per Intermodal Transport Unit (ITU) for all terminals in Italy. As far as non-Italian terminals are concerned, some terminal operators in Belgium, The Netherlands and Germany have been interviewed: they have provided similar values, equal to around 35 € per ITU;
- $n_{ITU}$ = number of intermodal transport units (ITUs) transported on each train;
- the cost of handling a train at railway terminals is multiplied by 2 because two transshipment movements have been considered: at the two intermodal centres of origin and destination of the railway path;
- $T(V_A, i, R_c)$ = electric traction cost [€/km]: it has been determined from the power consumption, in kWh/km, multiplied by the cost of electricity, in €/kWh. The power consumption has been calculated considering all resistances to motion. Details on the calculation of the power consumption are provided in the following. Only electrified lines have been considered. In Europe, usually, non-electrified lines show bad geometrical characteristics, particularly high gradients and sharp horizontal curves. Therefore, the diesel traction cost has not been taken into account in our research. The traction cost is a function of: the speed in rank A of freight trains on the link ($V_A$); the link grade ($i$); the curvature resistance ($R_c$). The resistances to motion have been calculated from the speed in rank A. The speed in rank A, in Italy, is the maximum speed for freight trains as reported in Mayer [14] (p. 49). The inertial resistance has been neglected, the traction is calculated at regime: acceleration and deceleration transitories have been neglected. Freight trains do not make scheduled intermediate stops from the origin to the destination, but, sometimes, they make some stops to let faster trains pass. We have taken into account of all these issues by taking the resistances to motion dependent on the speed in rank A, $V_A$, instead of the average speed, $V_M$.

**Table 1.** Comparison of the proposed cost function with the similar cost functions existing in literature: namely those proposed by Grosso [5] and Baumgartner [6].

| Component | Grosso | Baumgartner | The Proposed Cost Function |
|---|---|---|---|
| Staff cost | Maximum, average, minimum values provided | Not taken into account in the cost function | Two reference values for the driver cost per hour were found in literature: in Italy [9] and Germany [8]. An average of these two values (38.5 €/h) has been assumed. Two drivers are necessary to operate a freight train in Italy, while only one in the rest of Europe. |
| Number of locomotives | Not explicitly calculated | Not explicitly calculated | Calculated in detail. The number of locomotives depends on the grade and curve resistances of each line section and has been determined, for each rail line, from the 'Operating Rules' ('Norme di Esercizio') [15]. The operating rules used by the rail transport companies, which effectively operate the services, have been assumed. |

**Table 1.** *Cont.*

| Component | Grosso | Baumgartner | The Proposed Cost Function |
|---|---|---|---|
| Amortization/rental/leasing cost of a locomotive | Maximum, average, minimum values provided | Calculated from an average purchase cost of a locomotive | In [8], an amortization cost in €/year, valid for locomotives specifically used for freight transport, is provided. In order to calculate the amortization cost in €/km, the number of km/year travelled by a locomotive for freight transport has been provided by Mercitalia Rail. |
| Maintenance cost of a locomotive | Maximum, average, minimum values provided | Calculated as a percentage of the amortization cost | Calculated as a percentage of the amortization cost as suggested by [8]. |
| Insurance cost of a locomotive | Maximum, average and minimum insurance costs are provided for the entire train and not for simply a locomotive | Not taken into account in the cost function | Calculated as a percentage of the amortization cost as suggested by [8]. |
| Amortization cost of a flat wagon | Maximum, average, minimum values provided | Calculated from an average purchase cost of a flat wagon | In [8] an amortization cost in €/year for a flat wagon is provided. In order to calculate the amortization cost in €/km, the km/year travelled by a flat wagon have been provided by Mercitalia Rail. |
| Maintenance cost of a flat wagon | Maximum, average, minimum values provided | Calculated as a percentage of the amortization cost | Calculated as a percentage of the amortization cost as suggested by [8]. |
| Insurance cost of a flat wagon | Maximum, average, minimum values provided | Not taken into account in the cost function | Calculated as a percentage of the amortization cost as suggested by [8]. |
| Handling cost at terminals | Maximum, average, minimum values provided | Not taken into account in the cost function | Calculated according to the costs provided by Terminali Italia [13] and by northern European terminals, as €/load unit. The handling cost at an Italian terminal is 32.5 €/load unit and between 30 and 35 €/ITU in the rest of Europe. The total number of ITUs (Intermodal Transport Units) to be considered for each train has been collected from interviews to the main MTOs operating between Italian and northern European terminals. |
| Rail track cost | Maximum, average, minimum values provided | Not taken into account in the cost function | Rail track costs, in €/km, have been collected for each rail line. Rail track costs are not only different from a country to another, but often also from a line to another in the same country. |
| Traction cost | Maximum, average, minimum values provided | Reference values have been provided for different values of line slope | It has been determined from the power consumption, in kWh/km, multiplied by the cost of electricity, in €/KWh. The power consumption has been calculated considering all resistances to motion on each line section. |

### 2.1.3. Calculation of the Traction Cost

As stated before, the electric traction cost, $T(V_A, i, R_c)$ [€/km], has been determined from the power consumption, in kWh/km, multiplied by the cost of electricity, in €/kWh.

The power consumption has been calculated basing on all resistances to motion. The resistances to motion considered in the calculation, as suggested in Micucci and Mantecchini [16], are the rolling resistance, the aerodynamic resistance, as well as the grade and curve resistances. In the calculation of these resistances, the speed in rank A, i.e., $V_A$, has been used. As stated before, we have taken into account the acceleration and deceleration transitories, by considering the resistances to motion dependent on the speed in rank A, $V_A$, instead of the average speed, $V_M$. The resistances were determined according to the methodology proposed in Vicuna [17], but the formulas for resistances, which were old, have been updated.

The rolling resistance has been calculated according to Szanto [18] (p. 2). The air resistance has been calculated according to Lai et al. [19] (p. 823). The grade resistance has been calculated

considering the slope of each line section in detail while the curve resistance has been calculated from the classic Von Rockl formula.

The resistances to motion depend on the weight of the train (locomotive + wagons). An E189 locomotive (produced by Siemens) has been considered, which has a weight of 87 tons [20]. This type of locomotive is currently used, by the rail company 'Rail Traction Company', in the international freight transport across the Alpine Passes of Brenner and Tarvisio. It is widely used in Europe because it is a multi-tension locomotive.

One of the most common flat wagons, for the transport of containers, is the Sgns, with an unladen weight of 17.5 t/wagon (source: Mercitalia Rail [21]). In order to determine the average number of wagons composing a train, and the average number of TEUs (or ITUs) loaded on each train, four main MTOs (Hupac, Mercitalia Intermodal, Kombiverkehr, and Lineas Intermodal) operating between Italy and northern Europe, have been interviewed.

The calculation of the weight of the train is described in detail in Lupi et al. [22]. The towed weight has resulted equal to 1234 tons. The total weight of the train has resulted 1321 tons if only one locomotive is used, 1408 tons if two locomotives are used, 1495 tons in case of three locomotives.

For the cost of electricity, the average prices, in €/kWh, applied to companies (companies in general, not railway companies in particular), in each European country, were taken into account. For example, in Italy the average price for electricity, in the first half of 2019, for companies, has been around 0.0952 €/KWh (source: Eurostat [23]). A different electricity price for each European country has been considered, taken from Eurostat [23].

### 2.1.4. Maximum Towable Weight on a Railway Line Section and 'Lines with Special Operation Characteristics'

There are two main constraints related to the maximum towable weight on a railway line:

(1)　The maximum towable weight due to the resistance of train couplers.

It depends on the geometrical characteristics of the rail line, in particular on the sum of the grade and curve resistances. The towed weight of the train considered in this study is 1234 tons. From RFI (Rete Ferroviaria Italiana, the Italian rail network manager) prescriptions (source: Prefazione Generale all'Orario di Servizio [24]), 20 N/kN is the maximum value for the sum of grade and curve resistances, due to the resistance of train couplers for a 1234 tons towable weight.

Indeed, the Disposition n° 18 of 19 November 2015, published by RFI [25], has removed the limit about the maximum towable weight due to the resistance of train couplers, in order to satisfy the requirements of freight railway companies which aim at improving their productivity operating longer trains. The disposition specifies that the maximum towable weight can be determined by the railway companies according to specific analyses performed by the companies themselves.

(2)　The maximum towable weight due to the maximum tractive effort of the chosen locomotive.

It depends on the type of locomotive used, but also on the geometry of the line. Each typology of locomotive can tow a different weight: for example, the E652 (six axles) [26] is capable of towing a greater weight than the E189 (four axles) on the same line. The maximum weight that a locomotive can tow on each section of a line is reported in the 'Operating Rules' ('Norme di Esercizio') [15]. This type of document is publicly available online only in Italy. For a towed weight of 1234 tons and an E189 locomotive, the maximum sum of grade and curve resistances allowable for one locomotive is 12 N/kN. If the sum of grade and curve resistances is greater, more than one locomotive is necessary.

Therefore, in our model:

- if the sum of the grade and curve resistances is less than or equal to 12 N/kN, only one locomotive has been used;
- if the sum of the grade and curve resistances is more than 12 and less than 20 N/kN, two locomotives (E189) have been used;

- if the limit of 20 N/kN is overcome on a secondary line (for example the Savona–Altare or the Parma–La Spezia lines), this line has not been included in the modelled rail network;
- if the limit of 20 N/kN is overcome on a main line (for example, the main lines crossing the Alps, such as the Brennero, the Frejus and the Semmering lines), information has been collected, from freight railway companies which operate along the given Alpine line, about how the trains are practically operated: in the following such lines will be called 'line with special operation characteristics'. In our model, on 'line with special operation characteristics', we consider the same number of locomotives as used by the MTOs in the real operations to tow a weight of 1234 tons.

## 2.2. The Calculation of Travel Times

The travel time in each line section is calculated basing on the RFI formula for the average speed $V_m$:

$$V_m = 0.60231 \cdot V_A \ [\text{km/h}] \tag{2}$$

where $V_A$ is the speed in rank A: the maximum speed for freight trains.

The speed value for rank A refers, generally, to short line sections (1–3 km long). But, in this research, much longer railway links, namely at least 15–20 km, have been taken into account. Therefore, the speed in rank A, on each railway link, has been taken equal to a weighted average of the speeds of all line sections included in the link, considering as weight the percentage of the length of the link with the given speed in rank A.

The values of the speed in rank A are publicly available only in Italy, and in a few other European countries, on the website of the rail infrastructure managers. However, in the other countries, the speed in rank A is not publicly available: only the speed in rank C, which is the speed for fast passenger trains, is provided in the websites of the rail infrastructure managers. Therefore, a formula has been set up, through linear regression analysis, basing on Italian data, which determines the speed in rank A given the speed in rank C:

$$V_A = 0.8636 \cdot V_C + 2.8732 \ [\text{km/h}] \tag{3}$$

The Equation (3) has been used for data of the other European countries, for which the speed values for fast passenger trains (which could be assimilated to the Italian rank C) were publicly available.

## 2.3. Generalized Cost of Rail Links

As stated in Lupi et al. [27], after interviews to experts in the field, monetary costs and travel times are the variables mostly taken into account by carriers and shippers in their transport mode choice. For modelling a multimodal freight transport network, the generalized cost can be determined as follows:

$$C_g = C_m + VOT \cdot t \tag{4}$$

where:

- $C_g$ = generalized cost (€),
- $C_m$ = monetary cost (€),
- $VOT$ = value of time (€/h),
- $t$ = time (h).

A high variability of $VOT$ has been observed in the literature. Consequently, in the analysis carried out in this paper, travel times and monetary costs have been considered separately. However, in this paper also generalized costs have been taken into account. The $VOT$ considered, for the calculation of generalized costs, is 0.96 €/(t·h), proposed by De Jong in 2004 [28]. We chose this $VOT$ as it is an average one, and we think it is the most reliable among those proposed in the literature.

## 3. Construction of the European Rail Network Model and Data Collection

As reported in the introduction, in order to evaluate the usage of Alpine passes, a great part of European rail network has been modelled (Figure 1): two scenarios were taken into account (see Figures 2 and 3):

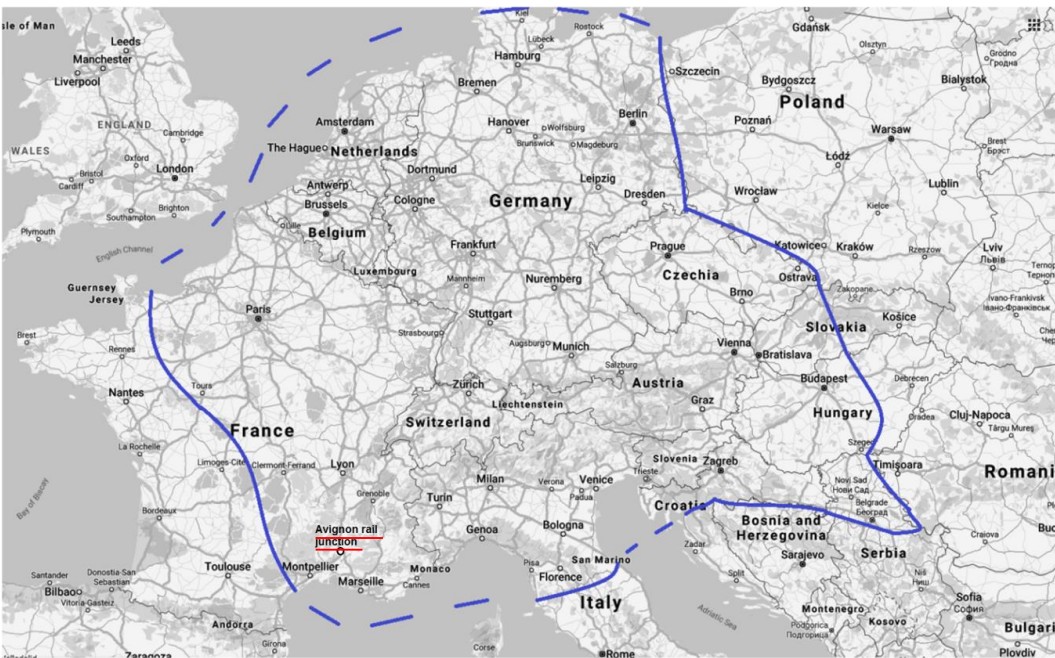

**Figure 1.** The European area taken into account in this study, circled in dark blue (the Avignon rail junction, mentioned several times in the paper, has been underlined in red).

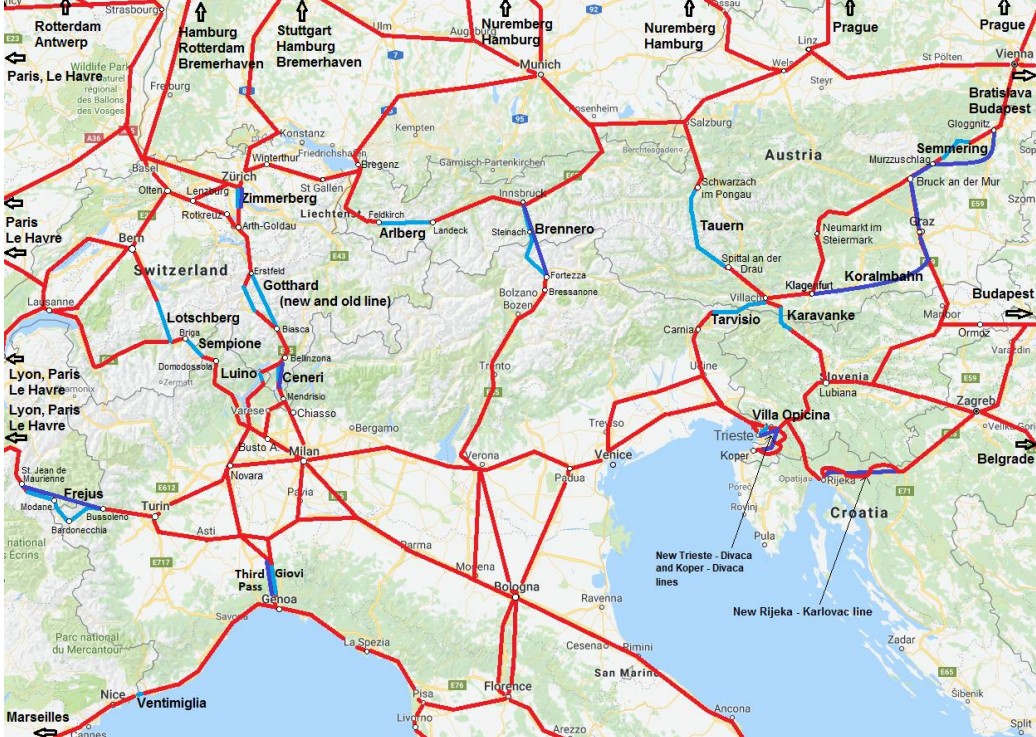

**Figure 2.** Alpine passes taken into account in this study. Alpine railway lines currently in operation ('current scenario') are represented in light blue; Alpine railway lines under construction or planned ('project scenario') are represented in dark blue.

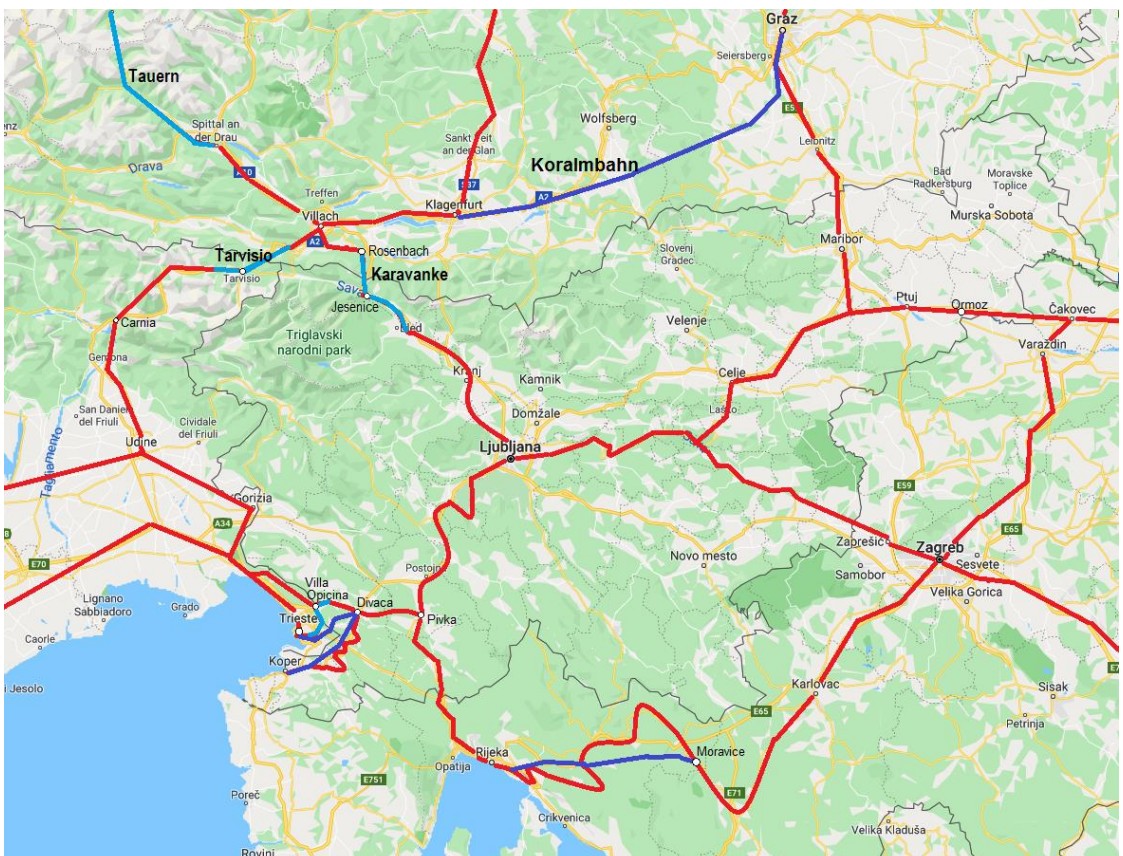

**Figure 3.** Enlargement of Figure 2. Railway lines currently in operation ('current scenario') are represented in red, and railway lines across Alpine Passes are represented in light blue; Alpine railway lines under construction or planned ('project scenario') are represented in dark blue.

- *Current scenario*

It involves only Alpine railway lines currently in operation: Ventimiglia line, Frejus line, Sempione line, Lötschberg base tunnel, Gotthard base tunnel, Ceneri line, Luino line, Arlberg line, Brenner line, Tauern line, Tarvisio line, Semmering line, Karavanke line. On the eastern part of the alps (Figure 3): Trieste–Divaca, Koper–Divaca, Rijeka–Divaca, and Rijeka–Zagreb lines.

- *Project scenario*

It involves also Alpine rail lines under construction or planned: the new Frejus base tunnel, the Ceneri base tunnel, the Brenner base tunnel, the new Semmering line (which includes: the 'Koralmbahn', that is the Klagenfurt—Graz new line, the new Semmering base tunnel, and the improvement of the existing lines: Graz—Bruck an der Mur and Bruck an der Mur–Murzzuschlag). The 'project scenario' considers also, on the east part of the Alps (Figure 3): the new lines Trieste—Divaca, Koper–Divaca, and Rijeka–Moravice (part of the Rijeka–Zagreb line). Furthermore, the 'project scenario' considers also, on the north-west part of Italy (near Genoa, Figure 2): the 'Third pass of Giovi' across the Ligurian Apennines.

Detailed information, about the railway network model of the European area, shown in Figure 1, is reported in Lupi et al. [22]. Herewith, only a focus on rail lines across the Alps is provided.

*3.1. Detailed Characteristics of the Alpine Railway Lines Taken into Account in This Study*

3.1.1. Gotthard Line and Its Branches

With regard to the Gotthard line, thanks to the opening of the new Gotthard Base tunnel, 57 km long, between Bodio on the south and Erstfeld on the north, the maximum sum of grade and curve resistances, which occurs in the south ramp, has been reduced from 27 to 13 N/kN [29,30]. It must be underlined that 13 N/kN is not recorded in the Gotthard base tunnel, but in the line from Bellinzona to Biasca. Because we took into account only one type of locomotive, the E189, for the whole network, and always the towed weight of 1234 tons, double traction has been considered from Bellinzona to Biasca and single traction on the rest of the line. However, it can be noticed that with another type of locomotive, for example Siemens Vectron, it is possible to tow 1234 t with only one locomotive also between Bellinzona to Biasca and this choice is more convenient economically. However, in our model, we considered only one locomotive and we chose the E189 because it is the most commonly used on the alpine lines. In a future extension of the work we will provide for different locomotive (with different costs), for the different railway lines.

The Gotthard line branches out in three lines on the south side, and in two lines on the north side.

The three branches on the south side are: the line across the Ceneri pass to Chiasso, a branch of the previous line crossing the Ceneri pass, then Mendrisio and after Varese to reach Busto Arsizio—Gallarate, the main Italian freight Intermodal Terminal [31], and the line across Luino. The line across Luino is not steep, nor tortuous, but it is composed of a single track; however, no improvement projects are currently planned on it. The Ceneri Pass north ramp (south direction) is operated in triple traction, as the sum of grade and curve resistances is 26 N/kN, while the Ceneri Pass south ramp (north direction) is operated in double traction: the sum of grade and curve resistances is 21 N/kN (source: interviews to the MTO Hupac).

The Ceneri Base tunnel is under construction and it is expected to be operative in September 2020: it is 15.4 km long. In the 'project scenario', the line Chiasso–Bellinzona has a maximum sum of grade and curve resistances equal to 12 N/kN which occurs from Chiasso to the south entrance of the tunnel. Instead in the Bellinzona–Chiasso direction, the line has a maximum sum of grade and curve resistances equal to 10 N/kN which occurs in the new Ceneri base tunnel [29,30].

The two branches of the Gotthard line on the north side involve: the line across Rotkreuz, Lenzburg and Olten; the line across the Zimmerberg pass to Zurich. While the first branch does not show relevant slopes and is operated in single traction, the Zimmerberg branch is currently operated in double traction: thanks to the new Zimmerberg tunnel, also this branch will be operated in single traction, with a maximum sum of grade and curve resistances of 12 N/kN.

In synthesis, taking into account the Gotthard line and all its branches, in the 'current scenario' it is operated: in the north direction, in single traction from Milan and from Varese to Lugano, in double traction from Lugano to Biasca, and again in double traction from Zug (at the beginning of current Zimmerberg line) to Zurich, while the rest of the line is operated in single traction; in the south direction, in double traction from Zurich to Zug (Zimmerberg), in triple traction from Bellinzona to Riviera/Bironico (on the south of the old Ceneri tunnel), grade 28‰, and the rest of the line is operated in single traction.

In the 'project scenario', it has been considered: in the northern direction double traction from Bellinzona to Biasca and single traction in the rest of the line and on its branches; in the south direction single traction in the entire line. In a future extension of the work, we will take into account different locomotives (with different costs) for the different railway lines, in order to consider operating the entire line, in both directions, in single traction.

3.1.2. Brenner Line

On the Brenner line (on the border between Italy and Austria), the maximum sum of grade and curve resistances is equal to: 26 N/kN on the Italian side, from Bressanone to the Brenner Pass, 51 km;

28 N/kN on the Austrian side from Steinach to the Brenner Pass, 13 km. On the Italian side, from Bressanone to the Brenner pass (51 km), double traction (both locomotives pulling the wagons) is used; on the Austrian side, double traction (both locomotives pulling the wagons) from Innsbruck to Steinach (26 km) and triple traction (two locomotives pulling and one pushing) from Steinach to the Pass (13 km) is used. The information on the number of locomotives was taken from Zurlo [32] and Schmittner [33]. On the Brenner line, it is allowable: a maximum towable weight of 1500 tons on the Italian side and of 1560 tons on the Austrian side (as reported in Schmittner [33]).

The new Brenner base tunnel will be built between Innsbruck (on the north side) and Fortezza (on the south side), and it will comprise also a bypass of Innsbruck. The line will be 55 km long excluding the bypass, and 64 km comprising the bypass; the maximum sum of grade and curve resistances in the tunnel will be 7 N/kN ([32,33]). This new line will be operated in single traction between Fortezza and Innsbruck, but in double traction from Bolzano to Fortezza. Also, a new line between Bolzano and Fortezza is planned; but, in this study, in the 'project scenario', only the Brenner base tunnel was taken into account, and not the new line Bolzano–Fortezza. The new line Bolzano—Fortezza was not considered in the 'project scenario', because the construction of this line is not foreseen in a short period, but it may be considered in a future extension of the study. Moreover, it has not been decided yet whether to construct only this line (plus a new high speed line Innsbruck–Kufstein to relieve the traffic on the existing line, which is very congested) or an entire new line between Verona and Munich, a part of which is the Brenner base tunnel [34].

### 3.1.3. Turin—Lyon 'Frejus' Line

On the Frejus line, on the Italian side, the maximum sum of grade and curve resistances is equal to 28 N/kN from Bussoleno to Salbertrand (22 km). From Salbertrand to Bardonecchia, the sum of grade and curve resistances varies and in particular it is equal to: 16 N/kN from Salbertrand to Oulx (6 km), 19.8 N/kN from Oulx to Beaulard (6 km) and 21.9 N/kN from Beaulard to Bardonecchia (5 km). Instead the sum of grade and curve resistances is equal to 31 N/kN, for only 3 km, between Bardonecchia and the beginning (on the Italian side) of the Frejus tunnel. On the French side, between Modane and the beginning of the Frejus tunnel, that is for 4 km, the maximum sum of grade and curve resistances is 31 N/kN; inside the tunnel, the maximum sum of grade and curve resistance is 31 N/kN for 7 km, from the beginning of the tunnel (on the French side) to the Italy/France border, while the rest of the tunnel (between the Italy/French border and the beginning of the tunnel on the Italian side), that is for 6.3 km, is flat. Between Modane (France) and Bussoleno (Italy), and vice versa, it is allowed: a maximum towable weight of 1150 t with double traction, and of 1600 t with triple traction (Osservatorio [35], in Ferrari [36]). The line on the French side, from St. Jean de Maurienne to Modane, shows a maximum sum of grade and curve resistances equal to 22 N/kN: this part of the line is operated with double traction with a maximum towable weight of 1600 t (source: interviews with Novatrans, one of the most important French MTOs, which operates on this line).

The new Turin–Lyon line is composed of [35]:

- The new Frejus base tunnel (sometimes called also Moncenisio base tunnel), between Bussoleno and St. Jean De Maurienne, 57.5 km long, of which 45 in France and 12.5 in Italy, with a maximum sum of grade and curve resistances of 13 N/kN (which does not take place in the tunnel itself but in the two, short, adduction ramps on both side of the tunnel). This tunnel will be operated in single traction by the majority of MTOs, but given the locomotive taken into account in this paper, double traction has been considered.
- A new line connecting the freight village of Turin/Orbassano with Bussoleno: this line will be constructed in order to shorten the path between these two places;
- A new line connecting S. Jean de Maurienne with Avressieux, and the duplication of the existing line between Avressieux and Saint Andrè Le Gaz: this line will also shorten considerably the current path, which follows the valley floors therefore it is quite long.

In the 'project scenario', however, only the Frejus base tunnel was taken into account, and not the new lines on the Italian side and on the French side. In a future step of this research, the entire new Brenner and Frejus lines will be taken into account.

### 3.1.4. Sempione and Lötschberg Lines

As far as the Sempione line is concerned, the section with a sum of grade and curve resistances above 20 N/kN, and equal to 24 N/kN, is very short, only 2 km, close to Iselle station: the line is operated with double traction between Domodossola (Italy) and Brig (Switzerland) in both directions (source: SBB [37]).

As to the Lötschberg line, thanks to the new Lötschberg tunnel, the maximum sum of grade and curve resistances, which occurred on the north ramp (south direction), has been reduced from 29 to 14 N/KN [38]. The Lötschberg line, given the towed weight and the E189 locomotive, is operated in double traction in our model.

### 3.1.5. The Tarvisio, Semmering and Tauern Lines

As far as the Tarvisio line is concerned, it does not have particular problems as to the resistance of train couplers. The maximum sum of grade and curve resistances is 14.6 N/kN from Carnia to Tarvisio pass (west ramp) and 20 N/kN from Villach to Tarvisio pass (east ramp). This line is operated in double traction between Carnia and Villach, in both directions.

As regards the Semmering line, the situation is the following. Between Villach and Klagenfurt (38 km) the line is flat. From Klagenfurt to Neumarkt im Steiermark (70 km), the line has a maximum sum of grade and curve resistances of 18 N/kN, while from Neumarkt im Steiermark to Bruck an Der Mur (99 km) the line is descending. In the opposite direction, from Bruck an Der Mur to Neumarkt im Steiermark (99 km), the line has a maximum sum of grade and curve resistances of 16 N/kN. From Bruck an Der Mur to Murzzuschlag (27 km), the line has a maximum sum of grade and curve resistances of 11 N/kN. From Murzzuschlag to the Semmering Pass (13 km), the line has a maximum sum of grade and curve resistances of 18 N/kN. From Gloggnitz to the Semmering Pass (27 km) the line has a maximum grade resistance of 22.5 N/kN but in particular a maximum curve resistance of 5.5 N/kN: some curves even have a radius of 150 m, and the maximum sum of grade and curve resistances is 28 N/kN.

The line is operated in double traction from Villach to Gloggnitz; in the opposite direction, it is operated in triple traction from Gloggnitz to Murzzuschlag and double traction from Murzzuschlag to Villach [39]. The part of the line between Villach and Klagenfurt could be operated in single traction, but it is only 38 km long and it is preceded and followed by lines operated in double traction. As a result, also this line portion is operated in double traction.

Although the Semmering line shows geometrical problems only for 40 km, an entire new line from Klagenfurt and Gloggnitz is under construction. This choice has been made also because the current Semmering line does not cross the important city of Graz. The new Semmering line will be composed of the following parts:

- new Koralmbahn, between Klagenfurt and Graz;
- upgrading of the old line between Graz and Bruck an der Mur
- upgrading of the old line between Bruck an der Mur and Murzzuschlag
- Semmering base tunnel between Murzzuschlag and Gloggnitz.

The new Semmering line will be entirely operated in single traction in both directions. The maximum sum of grade and curve resistance, in the direction Villach—Vienna, which will take place from Graz to Bruck an der Mur, is equal to 11 N/kN. In the direction Vienna—Villach, the maximum sum of grade and curve resistances is 9 N/kN, which will take place in the Koralmbahn east ramp.

The Tauern Line connects Villach with Salzburg, but the steepest part is located between Spittal an der Drau (located on the south) and Schwarzach im Pongau (located on the north) in both directions;

this line involves also the Tauern tunnel, 8.37 km long. The line has been highly improved recently, in order to reduce the curves, but the slopes have remained high and the maximum sum of grade and curve resistances is 27 N/kN in both directions. The line is operated in triple traction between Spittal an der Drau and Shwarzach im Pongau in both directions and in double traction in the rest of the line in both directions.

### 3.1.6. The Lines Trieste–Divaca, Koper–Divaca, Rijeka–Pivka, and Rijeka–Zagreb

Other lines 'with special operation characteristics' are those running from the ports of Trieste, Koper and Rijeka to the internal Karst plateau. In particular (see Figure 3):

- Line Trieste–Divaca: The railway line from Trieste Campo Marzio to Villa Opicina (border Italy—Slovenia), 15 km, is a part of the line Trieste–Divaca; it shows a sum of grade and curve resistances of 25 N/kN. This railway is part of the Trieste—Ljubljana path and it is operated with triple traction from Trieste Campo Marzio to Villa Opicina (Source: interviews to Alpe Adria, the main MTO operating railway connections to/from the Trieste Campo Marzio rail terminal) and in single traction in the opposite direction. Between Villa Opicina and Ljubljana, instead, the sum of grade and curve resistances is below 20 N/kN and the line is operated with double traction in both directions.
- Line Koper–Divaca: A portion, of about 18 km, of the line from Koper to Divaca, shows a maximum sum of grade and curve resistances of 23 N/kN. This line is operated in double traction (source: interviews to Metrans, the main MTO operating on this line) from Koper to Divaca and single traction in the opposite direction. But, the second locomotive is not added at the beginning of the train but at the end of the train, in order to reduce the stress on train couplers. This line is part of the Koper—Ljubljana path.
- Line Rijeka–Pivka: A portion of 15 km, close to Rijeka port (from Rijeka to Jurdani), of the line Rijeka–Pivka, shows a sum of grade and curve resistances of 27 N/kN. This line is operated with triple traction in this line portion of 15 km, and with double traction in the rest of the line. In the opposite direction, i.e., from Pivka ro Rijeka, the entire line is operated in double traction (source: interviews to Metrans). This line is a part of the Rijeka—Ljubljana path.
- Line Rijeka–Zagreb: A portion of about 25 km, between Rijeka and Moravice, of the line from Rijeka to Zagreb, shows a maximum sum of grade and curve resistances of 28 N/kN. Considering the entire line from Rijeka to Zagreb: the line is operated with triple traction from Rijeka to Moravice and the rest of the line is operated with double traction; while in the opposite direction the entire line is operated with double traction (source: interviews to Metrans).

Some new lines are planned (Figure 3), but only the Koper–Divaca is under construction (since March 2019). All these new lines have been considered in the 'project scenario':

- New line Trieste–Divaca (Figure 3): several projects have been proposed, but none is approved yet. The new line will have a maximum slope of 17‰, and a maximum sum of grade and curve resistances of 18 N/kN. This line will be operated in double traction from Trieste to Divaca and single traction in the opposite direction. It is not possible to design a line with lower slopes because of geographical constraints due to the high degree of urbanization of the Trieste area.
- New line Koper–Divaca (Figure 3): a new line is under construction (since March 2019), which will be long 27 km, against the 39 km of the line currently in operation, and its sum of grade and curve resistances will be 10 N/kN. This line is strongly supported by the Slovenian government because it will improve the competitiveness of the port of Koper against that of Trieste.
- New line Rijeka–Zagreb: A new railway line between Rijeka and Moravice (Figure 3), which is part of the Rijeka–Zagreb line, is planned. The new line will have a maximum slope of 12.5‰ and a maximum sum of grade and curve resistances of 13 N/kN. It will be operated in double traction because the rail Moravice—Karlovac, which is also part of the Rijeka–Zagreb line, requires

double traction. However, an improvement of the entire line Rijeka–Zagreb, that is, also between Moravice and Zagreb, is planned, also because the great majority of this line has only one track. However, as this improvement is planned for a distant future, it was not taken into account in the 'project scenario'.

### 3.1.7. The Arlberg Line

The Arlberg line connects Innsbruck, located along the Brenner Line, with Bregenz, close to the border between Austria and Switzerland. This line is part of a path which connects Innsbruck with Zurich (which is located on a branch of the Gotthard Line). This line has a slope of 26‰ on the west ramp and of 31‰ on the east ramp; the sum of grade and curve resistances is 27 N/kN on the west ramp and of 32 N/kN on the east ramp. It is operated in triple traction in both directions between Landeck and Feldkirch and single traction in the rest of the line. Despite these bad geometrical characteristics, there are no plans to improve it, because it is currently used by local traffic.

### 3.1.8. The Karavanke Line

The Karavanke line connects Villach, in Austria, with Ljubljana. It is also called Jesenice line because it crosses this city. The north ramp (Austrian side) shows a maximum sum of grade and curve resistances of 21 N/kN, while the south ramp (Slovenian side) has a maximum sum of grade and curve resistances of 30 N/kN. The line is operated in double traction in the south direction, that is from Villach to Ljubljana; instead in the north direction, it is operated in triple traction from Ljubljana to Rosenbach and in double traction from Rosenbach to Villach. Although this line shows bad geometric characteristics, there are currently no plans to improve it. However, thanks to the new Semmering line currently under construction, it will be possible to reach more easily Ljubljana from northern Europe through the line across Maribor, which is less steep and tortuous than the Karavanke line.

### 4. Calculation of Minimum Cost Paths between Pairs of Italian and European Terminals

In order to evaluate the importance of each Alpine pass, the optimal paths, between pairs of Italian, from one side, and European terminals, from the other side, have been determined. Clearly, only those routes which involve crossing an Alpine pass were taken into account. Because high variability of *VOT* has been observed in the literature, three different route optimizations have been carried out: by travel times, by monetary costs and by generalized costs. The best routes of minimum cost or of minimum travel time have been determined by means of a Dijkstra algorithm [40].

The terminals taken into account in the study have been the following:

- Italy: Turin (Orbassano freight village), Novara, Rivalta Scrivia, Busto Arsizio–Gallarate, Milan Smistamento, Parma, Verona, Bologna, Padua, Venice, Trieste, Genoa, La Spezia, Leghorn, Prato (Prato freight village is close to Florence).
- Germany: Munich, Stuttgart, Duisburg, Berlin, Hamburg, Bremerhaven.
- France: Le Havre, Paris, Lyon, Marseilles, Avignon rail junction (beginning of the rail corridor to Barcelona, Avignon rail junction is shown in Figure 1).
- Belgium and The Netherlands: Antwerp, Rotterdam.
- Austria and Switzerland: Vienna, Zurich and Basel.
- Central-Eastern Europe: Koper, Rijeka, Ljubljana, Zagreb, Belgrade, Budapest, Bratislava, Prague.

### 5. Results

The routes, between pairs of the terminals mentioned above, and crossing Alpine passes, have been determined. In this section, at first the usage of Alpine passes by these routes is determined (see Table 2). After, travel times, monetary costs, and generalized costs for the 'current' and the 'project' scenarios, are calculated and compared (see Table 3).

**Table 2.** Alpine Passes crossed by minimum travel time and cost routes in the 'current scenario' and in the 'project scenario'. Only some reference O–D pairs are reported (see Appendix A for a larger number of O–D pairs).

| O–D Pair | | Current Scenario | Project Scenario |
|---|---|---|---|
| Leghorn—Paris | Travel time | Giovi Pass, Frejus old tunnel | Third Pass of Giovi, New Frejus base tunnel |
| | Monetary cost | Giovi Pass, Frejus old tunnel | Third Pass of Giovi, New Frejus base tunnel |
| | Generalized cost | Giovi Pass, Frejus old tunnel | Third Pass of Giovi, New Frejus base tunnel |
| Milan—Avignon junction | Travel time | Giovi Pass, Ventimiglia | New Frejus base tunnel |
| | Monetary cost | Giovi Pass, Ventimiglia | New Frejus base tunnel |
| | Generalized cost | Giovi Pass, Ventimiglia | New Frejus base tunnel |
| Novara—Paris | Travel time | Luino, Gotthard base tunnel | New Frejus base tunnel |
| | Monetary cost | Luino, Gotthard base tunnel | New Frejus base tunnel |
| | Generalized cost | Luino, Gotthard base tunnel | New Frejus base tunnel |
| Milan—Paris | Travel time | Chiasso, Ceneri Pass, Gotthard base tunnel | Chiasso, New Ceneri base tunnel, Gotthard base tunnel |
| | Monetary cost | Chiasso, Ceneri Pass, Gotthard base tunnel | Chiasso, New Ceneri base tunnel, Gotthard base tunnel |
| | Generalized cost | Chiasso, Ceneri Pass, Gotthard base tunnel | Chiasso, New Ceneri base tunnel, Gotthard base tunnel |
| Novara—Rotterdam | Travel time | Luino, Gotthard base tunnel | Luino, Gotthard base tunnel |
| | Monetary cost | Luino, Gotthard base tunnel | Luino, Gotthard base tunnel |
| | Generalized cost | Luino, Gotthard base tunnel | Luino, Gotthard base tunnel |
| Milan—Duisburg | Travel time | Chiasso, Ceneri Pass, Gotthard base tunnel | Chiasso, New Ceneri base tunnel, Gotthard base tunnel |
| | Monetary cost | Chiasso, Ceneri Pass, Gotthard base tunnel | Chiasso, New Ceneri base tunnel, Gotthard base tunnel |
| | Generalized cost | Chiasso, Ceneri Pass, Gotthard base tunnel | Chiasso, New Ceneri base tunnel, Gotthard base tunnel |
| Milan—Hamburg | Travel time | Chiasso, Ceneri Pass, Gotthard base tunnel | Chiasso, New Ceneri base tunnel, Gotthard base tunnel |
| | Monetary cost | Chiasso, Ceneri Pass, Gotthard base tunnel | Chiasso, New Ceneri base tunnel, Gotthard base tunnel |
| | Generalized cost | Chiasso, Ceneri Pass, Gotthard base tunnel | Chiasso, New Ceneri base tunnel, Gotthard base tunnel |
| Milan—Berlin | Travel time | Chiasso, Ceneri Pass, Gotthard base tunnel | Chiasso, New Ceneri base tunnel, Gotthard base tunnel |
| | Monetary cost | Chiasso, Ceneri Pass, Gotthard base tunnel | Chiasso, New Ceneri base tunnel, Gotthard base tunnel |
| | Generalized cost | Chiasso, Ceneri Pass, Gotthard base tunnel | Chiasso, New Ceneri base tunnel, Gotthard base tunnel |
| Milan—Munich | Travel time | Brenner Pass | Brenner base tunnel |
| | Monetary cost | Brenner Pass | Brenner base tunnel |
| | Generalized cost | Brenner Pass | Brenner base tunnel |

**Table 2.** *Cont.*

| O–D Pair | | Current Scenario | Project Scenario |
|---|---|---|---|
| Venice—Munich | Travel time | Tarvisio, Tauern | Brenner base tunnel |
| | Monetary cost | Tarvisio, Tauern | Brenner base tunnel |
| | Generalized cost | Tarvisio, Tauern | Brenner base tunnel |
| Milan—Vienna | Travel time | Tarvisio, Klagenfurt—Bruck old line, Semmering Pass | Tarvisio, Koralmbahn, Semmering base tunnel |
| | Monetary cost | Tarvisio, Klagenfurt—Bruck old line, Semmering Pass | Tarvisio, Koralmbahn, Semmering base tunnel |
| | Generalized cost | Tarvisio, Klagenfurt—Bruck old line, Semmering Pass | Tarvisio, Koralmbahn, Semmering base tunnel |
| Trieste—Vienna | Travel time | Tarvisio, Klagenfurt-Bruck old line, Semmering Pass | Tarvisio, Koralmbahn, Semmering base tunnel |
| | Monetary cost | Villa Opicina, Divaca, Ljubljana, Maribor, Semmering Pass | New Trieste—Divaca line, Ljubljana, Maribor, Semmering base tunnel |
| | Generalized cost | Tarvisio, Klagenfurt-Bruck old line, Semmering Pass | Tarvisio, Koralmbahn, Semmering base tunnel |
| Koper—Vienna | Travel time | Koper-Divaca old line, Villa Opicina, Tarvisio, Klagenfurt—Bruck old line, Semmering Pass | Koper-Divaca new line, Villa Opicina, Tarvisio, Koralmbahn, Semmering base tunnel |
| | Monetary cost | Koper—Divaca old line, Ljubljana, Maribor, Semmering Pass | Koper-Divaca new line, Ljubljana, Maribor, Semmering base tunnel |
| | Generalized cost | Koper—Divaca old line, Ljubljana, Maribor, Semmering Pass | Koper—Divaca new line, Ljubljana, Maribor, Semmering base tunnel |
| Vienna—Lyon | Travel time | Not through the Alps but via Munich, Bregenz, Zurich, Bern, Lausanne | Semmering base tunnel, Koralmbahn, Tarvisio, new Frejus base tunnel |
| | Monetary cost | Semmering Pass, Bruck—Klagenfurt old line, Tarvisio, Frejus old tunnel | Semmering base tunnel, Koralmbahn, Tarvisio, new Frejus base tunnel |
| | Generalized cost | Not through the Alps but via Munich, Bregenz, Zurich, Bern, Lausanne | Semmering base tunnel, Koralmbahn, Tarvisio, new Frejus base tunnel |
| Munich—Avignon rail junction | Travel time | Not through the Alps but via Bregenz, Zurich, Bern, Lausanne | Brenner base tunnel, new Frejus base tunnel |
| | Monetary cost | Not through the Alps but via Bregenz, Zurich, Bern, Lausanne | Brenner base tunnel, new Frejus base tunnel |
| | Generalized cost | Not through the Alps but via Bregenz, Zurich, Bern, Lausanne | Brenner base tunnel, new Frejus base tunnel |
| Munich—Marseilles | Travel time | Brenner Pass, Giovi Pass, Ventimiglia | Brenner base tunnel, Third Pass of Giovi, Ventimiglia |
| | Monetary cost | Brenner Pass, Giovi Pass, Ventimiglia | Brenner base tunnel, Third Pass of Giovi, Ventimiglia |
| | Generalized cost | Brenner Pass, Giovi Pass, Ventimiglia | Brenner base tunnel, Third Pass of Giovi, Ventimiglia |

*5.1. Usage of Alpine Passes*

The number of O–D pairs taken into account in this study is very large. Consequently, in the Table 2, only the results for a selection of some representative O–D pairs are reported. The results for a larger part of the O–D pairs, taken into account in this study, are reported in the Appendix A.

If the origin or destination is a port, the rail terminal closest to the maritime container terminal was taken into account: for example, PSA Genova Prà, La Spezia Marittima, Hamburg Eurogate/Waltershof, Rotterdam Euromax. If the origin or destination is an inland city, the most important rail freight terminal close to the city centre was taken into account, for example: Turin Orbassano, Paris Noisy, München Riem, Wien Süd.

The paths of minimum travel time have been determined from the optimization by travel times, the paths of minimum monetary cost have been determined from the optimization by monetary costs, while the paths of minimum generalized cost have been determined from the optimization by generalized costs.

In the 'project scenario', the new lines will take up the demand of the O–D pairs that in the 'current scenario' use the corresponding old railway lines, but also the demand of other O–D pairs that in the 'current scenario' use other passes. For example, the new Brenner base tunnel, in the 'project scenario', will take up the demand of the O–D pair Milan–Munich (that in the 'current scenario' uses the old line through Brenner pass), but also the demand of the O–D pair Venice–Munich which currently crosses the Tarvisio and Tauern rail lines. Similarly, the Frejus base tunnel will take up the demand of the O–D pair Genoa–Paris (that currently uses the old line), but also the demand of the O–D pair Novara–Paris which currently crosses the Luino line and the Gotthard base tunnel.

The most important railway axis, in the European region considered, is the Gotthard one (see also the larger set of the O–D pairs considered in the Appendix A), especially thanks to the construction of the base tunnel, already in operation in the 'current scenario'. But, the construction of the Brenner base tunnel will be able to take up a quota of the demand currently crossing the Gotthard line.

In addition, the Brenner base tunnel will also take up a quota of demand currently using the Tarvisio and Tauern lines. The Frejus base tunnel, in the Turin–Lyon line, will also be able to take up a quota of the demand to/from the western part of the Padan Plain, especially because the rail track cost is much lower in France and Italy than in Switzerland. The Frejus base tunnel will also be able to attract a quota of demand that currently crosses the Ventimiglia line: indeed, currently, the best path between Milan and the Avignon rail junction crosses Ventimiglia, while in the 'project scenario' it crosses the Frejus base tunnel. The Avignon rail junction is very important because it is the beginning of the line to Barcelona, on the European Mediterranean Corridor. It consists of the junction between the Lyon–Marseilles rail line and the rail line to Barcelona and is located about 3 km on the west of Avignon, as shown in Figure 1. This junction is taken into account because Barcelona is outside the area modelled in this study (see Figure 1).

The new Semmering line, instead, will be very important for Central-Eastern European destinations: in particular, between Zagreb and southern Germany, currently the best path crosses the Karavanke and Tauern lines, while in the 'project scenario' it crosses the line across Maribor and the Semmering base tunnel. This shift of the demand is very positive as the Karavanke line shows bad geometrical characteristics. This will also improve the competitiveness of Adriatic ports, against northern European ports, to reach Central-Eastern European destinations. Indeed, Adriatic ports are in a more favourable position, than northern European ports, for the maritime route to Far East.

Finally, after the construction of the new lines, the east—west path across northern Italy, through the Padan Plain, will become more convenient. This can be observed, for example, for the O–D pairs Munich–Avignon junction and Vienna–Lyon. The alternative route, on the north of the Alps, is much longer and, partially, involves very old lines. The path crosses Munich, Bregenz, Winterthur, Zurich, Bern and Lausanne. The route Munich–Bregenz involves old lines, built in the middle of the XIX century, with poor geometrical characteristics. The path between Bregenz and Lausanne is composed of several short portions of many different lines: this route generally shows good geometrical characteristics, but it

is quite long. Instead the alternative route, across northern Italy through the Padan Plain, in the 'project scenario', will be made of shorter paths with good geometrical characteristics. The competitiveness of the alternative route across northern Italy through the Padan Plain will be further improved by the construction of: the new line Bolzano—Fortezza (37 km), on the south side of the Brenner base tunnel; the new lines Turin/Orbassano–Bussoleno (30 km) and S. Jean de Maurienne–Avressieux (74 km) and the duplication of the existing line between Avressieux and St. André Le Gaz (18 km), all lines located along the Turin–Lyon route path.

*5.2. Travel Times, Monetary Costs and Generalized Costs in the Current and in the Project Scenarios*

In this section, travel times, monetary costs and generalized costs, between the O–D pairs taken into account in Section 5.1, are reported.

In Table 3, it is shown that, in absolute value, significant decreases, of travel times, monetary costs and generalized costs, can be observed, from the 'current scenario' to the 'project scenario', but in percentage terms, these decreases are not high. This occurs because the distances between origin/destination pairs accounted for are usually very high, while the lines across Alpine passes are not so long. For example, the distance Milan–Munich is about 550 km, while the Brenner base tunnel will be 55 km long. In Table 3, travel times, monetary costs and generalized costs, are reported for the selection of O–D pairs already considered in Table 2. The same information, for the larger set of O–D pairs already taken into account in the Appendix A, is reported in the Appendix B.

**Table 3.** Travel times [h], monetary costs [€/train] and generalized costs [€/train] between selected O–D pairs. (see the Appendix B for a larger number of O–D pairs).

| O–D Pair | Travel Time (h) | | Monetary Cost (€/train) | | Generalized Cost (€/train) | |
|---|---|---|---|---|---|---|
| | Current Scenario | Project Scenario | Current Scenario | Project Scenario | Current Scenario | Project Scenario |
| Leghorn—Paris | 16.30 | 15.16 | 20,398.3 | 19,323.6 | 31,431.5 | 29,577.5 |
| Milan—Avignon rail junction | 8.97 | 8.17 | 13,186.1 | 12,420.4 | 19,236.5 | 17,931.5 |
| Novara—Paris | 11.99 | 11.54 | 16,938.9 | 16,173.2 | 25,014.5 | 23,929.5 |
| Milan—Paris | 12.18 | 11.98 | 17,692.9 | 17,080.2 | 25,773.5 | 25,067.5 |
| Novara—Rotterdam | 15.71 | 15.71 | 20,933.4 | 20,933.4 | 32,179.5 | 32,179.5 |
| Milan—Duisburg | 12.81 | 12.61 | 20,290.5 | 19,677.7 | 28,590.5 | 27,884.5 |
| Milan—Hamburg | 17.28 | 17.08 | 26,254.0 | 25,641.2 | 37,430.5 | 36,724.5 |
| Milan—Berlin | 16.59 | 16.22 | 25,060.2 | 24,130.1 | 36,248.5 | 34,679.5 |
| Milan—Munich | 8.30 | 7.36 | 13,549.5 | 12,619.5 | 19,032.5 | 17,463.5 |
| Venice—Munich | 7.53 | 7.08 | 13,088.2 | 12,158.2 | 18,257.5 | 16,829.5 |
| Milan—Vienna | 11.32 | 10.18 | 16,727.6 | 16,226.5 | 24,349.5 | 22,932.5 |
| Trieste—Vienna | 8.14 | 6.90 | 12,107.5 | 11,450.2 | 18,141.5 | 16,724.5 |
| Koper—Vienna | 9.69 | 8.49 | 12,340.6 | 11,401.2 | 19,667.5 | 17,379.5 |
| Vienna—Lyon | 18.08 | 16.89 | 24,347.8 | 23,046.3 | 36,996.5 | 34,244.5 |
| Munich—Avignon rail junction | 15.93 | 15.45 | 21,636.4 | 20,834.4 | 32,675.5 | 31,146.5 |
| Munich—Marseilles | 15.96 | 14.78 | 20,882.5 | 19,764.5 | 31,757.5 | 29,832.5 |

The greatest reductions of travel times, monetary costs and generalized costs, have been observed for the O–D pairs whose best paths in the 'project scenario' cross the new Semmering or the new Frejus base tunnels; that is the following O–D pairs: Milan–Vienna, Trieste–Vienna, Koper—Vienna (new Semmering); Vienna–Lyon, Genoa–Paris (new Frejus). This occurs because the Frejus and Semmering old lines show very poor geometrical characteristics. Another O–D pair which registers a significant decrease of travel times and costs is Munich–Marseilles, thanks to the new Third Pass of Giovi and the Brenner base tunnel.

However, in this study, shunting times and monetary costs, that is the times and monetary costs for adding and removing locomotives when double and triple traction is required, have been neglected. This may be a remarkable undervalue of the 'project scenario'. In a future work, these times and monetary costs will also be considered in the cost function.

## 6. Discussion and Conclusions

In this paper, the usage of Alpine passes is analyzed, before: 'current scenario', and after: 'project scenario', the construction of the new base tunnels and the new railway lines under construction or planned. The rail network of a large part of Europe has been modelled through a graph and a new cost function for rail links has been developed.

Only a few rail link cost functions exist in the literature and, generally, they are not very detailed: only the cost functions proposed by Grosso [5], Baumgartner [6] and Dolinayovà [7] take into account in detail the monetary cost components.

However, the cost function proposed by Grosso does not always provide precise values. For example, the number of locomotives is not taken into account explicitly, while for the traction cost only a maximum, a minimum and an average value are provided, which do not depend explicitly on the geometrical characteristics of rail lines.

The cost function proposed by Baumgartner provides more precise values, especially as far as locomotive and wagon costs are concerned. However, some cost components are not taken into account: the staff cost, and the rail track cost. Moreover, as regards the traction cost, only some reference values have been provided for some different line slopes.

The cost function proposed by Dolinayovà [7] is more complete, but it does not consider explicitly the influence of the geometrical characteristics of railway lines on the traction cost and on the number of locomotives necessary to operate a train. Moreover, it does not report reference values for all cost components taken into account, but only for some of them.

The new proposed cost function takes into account: staff cost; amortization, maintenance and insurance costs of locomotives and wagons; rail track usage cost; traction cost. The number of locomotives is taken into account explicitly, and the traction cost has been determined precisely given all the resistances to motion. Detailed information on the geometry of each rail line has been collected, with special concern for the lines crossing the Alps. For each Alpine line, the number of locomotives necessary to operate a train, and the maximum towed weight have been collected.

The proposed cost function will be improved as follows. Firstly, times and monetary costs of shunting operations, related to adding and removing locomotives in case of double and triple traction, were not considered in the proposed cost function, and will be taken into account in a future step of this research. Secondly, the same locomotive is taken into account in the whole network, but in some lines, this results in a greater number of shunting operations. Therefore, in a future step of the research, a different locomotive will be considered on each line.

From the analysis carried out, as far as the 'current scenario' is concerned, the following observations can be done.

The most important rail line is the Gotthard one: because of its geometrical characteristics, as the Gotthard base tunnel has been recently opened, and because of its geographical position, as it is connected to all important destinations, not only in Switzerland, but also in Germany, France, Belgium, and The Netherlands. Moreover, the three branches of the Gotthard line, on the south of the base tunnel, across Luino, Varese, and Chiasso, connect the Gotthard to all destinations of the Italian Padan Plain.

The Brenner line is currently in competition with the Gotthard and Tarvisio-Tauern lines for origins in north Italy and destinations in Central-Eastern (for example Berlin) and Southern Germany (for example Munich). The current Brenner line is disadvantaged by its geometrical characteristics: indeed, double traction on the Italian side and triple traction on the Austrian side are necessary.

Other important railway lines, for Central-Eastern European destinations, are: Tarvisio, Tauri, Semmering and Karavanke. The Karavanke line is currently very used to connect southern Germany with Slovenia and Croatia, but it shows poor geometrical characteristics. But as regards the alternative paths: the line across the Semmering Pass is tortuous, while most lines across Hungary, although they are almost flat, are not electrified.

In the 'project scenario', the competitiveness of Brenner and Frejus lines will increase, after the construction of the new base tunnels, but also the Gotthard line will increase the competitiveness after

the construction of the new Ceneri and Zimmerberg base tunnels. As a result, in the 'project scenario', there is a strong competition among these three lines. The Frejus line, although the geographical position of the Gotthard line is more advantageous, will take benefit from the high rail track costs in Switzerland, which highly increase the monetary cost of the Gotthard line. This competition will decrease the importance of the Sempione–Lötschberg line: despite its good geometrical characteristics, this line is longer than the Gotthard and the new Frejus lines.

The new Semmering base tunnel and the Koralmbahn line will improve the accessibility by rail to Trieste and Koper from Central-Eastern European origins/destinations. The new Semmering base tunnel will improve the accessibility to Ljubljana, Zagreb, Belgrade, Koper and Rijeka from Central-Eastern European origins/destinations. Indeed, the Semmering base tunnel will increase the use of the path across Maribor, alternative to the Karavanke line, which shows poor geometrical characteristics.

The east–west path, bypassing the Alps to the north, between Munich and Bregenz, crosses very old lines with poor geometrical characteristics, while between Bregenz and Lausanne is composed of a large number of railway sections belonging to several different lines, and therefore it is not direct. The new Alpine lines will allow an east-west path across Italy, crossing two times the Alps, instead of bypassing the Alps on the north. Moreover, the new line Turin–Lyon is fundamental for the Mediterranean corridor, in alternative to the line across Ventimiglia, which is very crowded, and in some parts, it is still single track.

Finally, the travel times, and monetary and generalized costs, between some representative O–D pairs, were calculated. This study has shown that the greatest reductions of travel times and costs, from the 'current scenario' to the 'project scenario', are observed by the O–D pairs which are connected by paths crossing the new Frejus base tunnel or the new Semmering base tunnel: indeed the current Frejus and Semmering lines show very bad geometrical characteristics.

In a future step of the research, the entire new lines Bolzano–Innsbruck and Turin–Lyon, and not only the new Brenner and Frejus base tunnels, will be taken into account.

**Author Contributions:** Conceptualization, M.L.; methodology, A.F.; software, A.F.; validation, M.L., A.F., A.P. and D.C.; formal analysis, M.L., A.F. and A.P.; investigation, A.F.and D.C.; resources, A.F.and D.C.; data curation, M.L. and A.P.; writing—original draft preparation, M.L., A.F., A.P. and D.C.; writing—review and editing, M.L., A.F., A.P. and D.C.; supervision, M.L. and A.P.; project administration, A.F.; funding acquisition, M.L. and A.P. All authors have read and agreed to the published version of the manuscript.

**Funding:** This research was carried out within the framework of the research project 'LIVEUROP—Analysis of the impact of the construction of the new 'European Platform' on the hinterland of the port of Leghorn' funded by 'Fondazione Livorno'.

**Conflicts of Interest:** The authors declare no conflict of interest.

## Appendix A

**Table A1.** Alpine Passes crossed by minimum travel time and by minimum cost routes, in the 'current scenario' and in the 'project scenario'.

| O–D Pair | | Current Scenario | Project Scenario |
|---|---|---|---|
| Turin—Paris | Travel time | Frejus old tunnel | New Frejus base tunnel |
| | Monetary cost | Frejus old tunnel | New Frejus base tunnel |
| | Generalized cost | Frejus old tunnel | New Frejus base tunnel |
| Paris—Turin | Travel time | Frejus old tunnel | New Frejus base tunnel |
| | Monetary cost | Frejus old tunnel | New Frejus base tunnel |
| | Generalized cost | Frejus old tunnel | New Frejus base tunnel |

**Table A1.** *Cont*.

| O–D Pair | | Current Scenario | Project Scenario |
|---|---|---|---|
| Genoa—Paris | Travel time | Giovi Pass, Frejus old tunnel | Third Pass of Giovi, New Frejus base tunnel |
| | Monetary cost | Giovi Pass, Frejus old tunnel | Third Pass of Giovi, New Frejus base tunnel |
| | Generalized cost | Giovi Pass, Frejus old tunnel | Third Pass of Giovi, New Frejus base tunnel |
| Paris—Genoa | Travel time | Frejus old tunnel, Giovi Pass | New Frejus base tunnel, Third Pass of Giovi |
| | Monetary cost | Frejus old tunnel, Giovi Pass | New Frejus base tunnel, Third Pass of Giovi |
| | Generalized cost | Frejus old tunnel, Giovi Pass | New Frejus base tunnel, Third Pass of Giovi |
| Leghorn—Paris | Travel time | Giovi Pass, Frejus old tunnel | Third Pass of Giovi, New Frejus base tunnel |
| | Monetary cost | Giovi Pass, Frejus old tunnel | Third Pass of Giovi, New Frejus base tunnel |
| | Generalized cost | Giovi Pass, Frejus old tunnel | Third Pass of Giovi, New Frejus base tunnel |
| Paris—Leghorn | Travel time | Frejus old tunnel, Giovi Pass | New Frejus base tunnel, Third Pass of Giovi |
| | Monetary cost | Frejus old tunnel, Giovi Pass | New Frejus base tunnel, Third Pass of Giovi |
| | Generalized cost | Frejus old tunnel, Giovi Pass | New Frejus base tunnel, Third Pass of Giovi |
| Milan—Avignon Junction | Travel time | Giovi Pass, Ventimiglia | New Frejus base tunnel |
| | Monetary cost | Giovi Pass, Ventimiglia | New Frejus base tunnel |
| | Generalized cost | Giovi Pass, Ventimiglia | New Frejus base tunnel |
| Avignon junction—Milan | Travel time | Ventimiglia, Giovi Pass | New Frejus base tunnel |
| | Monetary cost | Ventimiglia, Giovi Pass | New Frejus base tunnel |
| | Generalized cost | Ventimiglia, Giovi Pass | New Frejus base tunnel |
| Turin—Rotterdam | Travel time | Sempione, Lötschberg | New Frejus base tunnel |
| | Monetary cost | Frejus old tunnel | New Frejus base tunnel |
| | Generalized cost | Frejus old tunnel | New Frejus base tunnel |
| Rotterdam—Turin | Travel time | Lötschberg, Sempione | New Frejus base tunnel |
| | Monetary cost | Frejus old tunnel | New Frejus base tunnel |
| | Generalized cost | Frejus old tunnel | New Frejus base tunnel |
| Novara—Paris | Travel time | Luino, Gotthard base tunnel | New Frejus base tunnel |
| | Monetary cost | Luino, Gotthard base tunnel | New Frejus base tunnel |
| | Generalized cost | Luino, Gotthard base tunnel | New Frejus base tunnel |
| Paris—Novara | Travel time | Gotthard base tunnel, Luino | New Frejus base tunnel |
| | Monetary cost | Gotthard base tunnel, Luino | New Frejus base tunnel |
| | Generalized cost | Gotthard base tunnel, Luino | New Frejus base tunnel |

**Table A1.** *Cont.*

| O–D Pair | | Current Scenario | Project Scenario |
|---|---|---|---|
| Novara—Antwerp/ Rotterdam | Travel time | Luino, Gotthard base tunnel | Luino, Gotthard base tunnel |
| | Monetary cost | Luino, Gotthard base tunnel | Luino, Gotthard base tunnel |
| | Generalized cost | Luino, Gotthard base tunnel | Luino, Gotthard base tunnel |
| Antwerp/ Rotterdam—Novara | Travel time | Gotthard base tunnel, Luino | Gotthard base tunnel, Luino |
| | Monetary cost | Gotthard base tunnel, Luino | Gotthard base tunnel, Luino |
| | Generalized cost | Gotthard base tunnel, Luino | Gotthard base tunnel, Luino |
| Turin—Hamburg | Travel time | Sempione, Lötschberg | Sempione, Lötschberg |
| | Monetary cost | Sempione, Lötschberg | Sempione, Lötschberg |
| | Generalized cost | Sempione, Lötschberg | Sempione, Lötschberg |
| Hamburg—Turin | Travel time | Lötschberg, Sempione | Lötschberg, Sempione |
| | Monetary cost | Lötschberg, Sempione | Lötschberg, Sempione |
| | Generalized cost | Lötschberg, Sempione | Lötschberg, Sempione |
| Milan—Paris | Travel time | Chiasso, Ceneri Pass, Gotthard Base tunnel | Chiasso, New Ceneri base tunnel, Gotthard base tunnel |
| | Monetary cost | Chiasso, Ceneri Pass, Gotthard Base tunnel | Chiasso, New Ceneri base tunnel, Gotthard base tunnel |
| | Generalized cost | Chiasso, Ceneri Pass, Gotthard Base tunnel | Chiasso, New Ceneri base tunnel, Gotthard base tunnel |
| Paris—Milan | Travel time | Gotthard Base Tunnel, Ceneri Pass, Chiasso | Gotthard Base Tunnel, New Ceneri base tunnel, Chiasso |
| | Monetary cost | Gotthard Base Tunnel, Ceneri Pass, Chiasso | Gotthard Base Tunnel, New Ceneri base tunnel, Chiasso |
| | Generalized cost | Gotthard Base Tunnel, Ceneri Pass, Chiasso | Gotthard Base Tunnel, New Ceneri base tunnel, Chiasso |
| Milan—Antwerp/ Rotterdam | Travel time | Chiasso, Ceneri Pass, Gotthard Base tunnel | Chiasso, New Ceneri base tunnel, Gotthard base tunnel |
| | Monetary cost | Chiasso, Ceneri Pass, Gotthard Base tunnel | Chiasso, New Ceneri base tunnel, Gotthard base tunnel |
| | Generalized cost | Chiasso, Ceneri Pass, Gotthard Base tunnel | Chiasso, New Ceneri base tunnel, Gotthard base tunnel |
| Antwerp/ Rotterdam—Milan | Travel time | Gotthard Base Tunnel, Ceneri Pass, Chiasso | Gotthard Base Tunnel, New Ceneri base tunnel, Chiasso |
| | Monetary cost | Gotthard Base Tunnel, Ceneri Pass, Chiasso | Gotthard Base Tunnel, New Ceneri base tunnel, Chiasso |
| | Generalized cost | Gotthard Base Tunnel, Ceneri Pass, Chiasso | Gotthard Base Tunnel, New Ceneri base tunnel, Chiasso |
| Milan—Duisburg | Travel time | Chiasso, Ceneri Pass, Gotthard Base tunnel | Chiasso, New Ceneri base tunnel, Gotthard base tunnel |
| | Monetary cost | Chiasso, Ceneri Pass, Gotthard Base tunnel | Chiasso, New Ceneri base tunnel, Gotthard base tunnel |
| | Generalized cost | Chiasso, Ceneri Pass, Gotthard Base tunnel | Chiasso, New Ceneri base tunnel, Gotthard base tunnel |

**Table A1.** *Cont.*

| O–D Pair | | Current Scenario | Project Scenario |
|---|---|---|---|
| Duisburg—Milan | Travel time | Gotthard Base Tunnel, Ceneri Pass, Chiasso | Gotthard Base Tunnel, New Ceneri base tunnel, Chiasso |
| | Monetary cost | Gotthard Base Tunnel, Ceneri Pass, Chiasso | Gotthard Base Tunnel, New Ceneri base tunnel, Chiasso |
| | Generalized cost | Gotthard Base Tunnel, Ceneri Pass, Chiasso | Gotthard Base Tunnel, New Ceneri base tunnel, Chiasso |
| Milan—Hamburg | Travel time | Chiasso, Ceneri Pass, Gotthard Base tunnel | Chiasso, New Ceneri base tunnel, Gotthard base tunnel |
| | Monetary cost | Chiasso, Ceneri Pass, Gotthard Base tunnel | Chiasso, New Ceneri base tunnel, Gotthard base tunnel |
| | Generalized cost | Chiasso, Ceneri Pass, Gotthard Base tunnel | Chiasso, New Ceneri base tunnel, Gotthard base tunnel |
| Hamburg/Milan | Travel time | Gotthard Base Tunnel, Ceneri Pass, Chiasso | Gotthard Base Tunnel, New Ceneri base tunnel, Chiasso |
| | Monetary cost | Gotthard Base Tunnel, Ceneri Pass, Chiasso | Gotthard Base Tunnel, New Ceneri base tunnel, Chiasso |
| | Generalized cost | Gotthard Base Tunnel, Ceneri Pass, Chiasso | Gotthard Base Tunnel, New Ceneri base tunnel, Chiasso |
| Milan—Berlin | Travel time | Chiasso, Ceneri Pass, Gotthard Base tunnel | Chiasso, New Ceneri base tunnel, Gotthard base tunnel |
| | Monetary cost | Chiasso, Ceneri Pass, Gotthard Base tunnel | Chiasso, New Ceneri base tunnel, Gotthard base tunnel |
| | Generalized cost | Chiasso, Ceneri Pass, Gotthard Base tunnel | Chiasso, New Ceneri base tunnel, Gotthard base tunnel |
| Berlin—Milan | Travel time | Gotthard Base Tunnel, Ceneri Pass, Chiasso | Gotthard Base Tunnel, New Ceneri base tunnel, Chiasso |
| | Monetary cost | Gotthard Base Tunnel, Ceneri Pass, Chiasso | Gotthard Base Tunnel, New Ceneri base tunnel, Chiasso |
| | Generalized cost | Gotthard Base Tunnel, Ceneri Pass, Chiasso | Gotthard Base Tunnel, New Ceneri base tunnel, Chiasso |
| Genoa—Zurich | Travel time | Giovi Pass, Luino, Gotthard base tunnel | Third Pass of Giovi, Luino, Gotthard base tunnel |
| | Monetary cost | Giovi Pass, Luino, Gotthard base tunnel | Third Pass of Giovi, Luino, Gotthard base tunnel |
| | Generalized cost | Giovi Pass, Luino, Gotthard base tunnel | Third Pass of Giovi, Luino, Gotthard base tunnel |
| Zurich—Genoa | Travel time | Gotthard base tunnel, Luino, Giovi Pass | Gotthard base tunnel, Luino, Third Pass of Giovi |
| | Monetary cost | Gotthard base tunnel, Luino, Giovi Pass | Gotthard base tunnel, Luino, Third Pass of Giovi |
| | Generalized cost | Gotthard base tunnel, Luino, Giovi Pass | Gotthard base tunnel, Luino, Third Pass of Giovi |
| Turin—Munich | Travel time | Luino, Gotthard base tunnel, Zimmerberg old line | Brenner base tunnel |
| | Monetary cost | Luino, Gotthard base tunnel, Zimmerberg old line | Brenner base tunnel |
| | Generalized cost | Luino, Gotthard base tunnel, Zimmerberg old line | Brenner base tunnel |

**Table A1.** *Cont*.

| O–D Pair | | Current Scenario | Project Scenario |
|---|---|---|---|
| Munich—Turin | Travel time | Zimmerberg old line, Gotthard base tunnel, Luino | Brenner base tunnel |
| | Monetary cost | Zimmerberg old line, Gotthard base tunnel, Luino | Brenner base tunnel |
| | Generalized cost | Zimmerberg old line, Gotthard base tunnel, Luino | Brenner base tunnel |
| Milan—Munich | Travel time | Brenner Pass | Brenner base tunnel |
| | Monetary cost | Brenner Pass | Brenner base tunnel |
| | Generalized cost | Brenner Pass | Brenner base tunnel |
| Munich—Milan | Travel time | Brenner Pass | Brenner base tunnel |
| | Monetary cost | Brenner Pass | Brenner base tunnel |
| | Generalized cost | Brenner Pass | Brenner base tunnel |
| Leghorn—Berlin/ Munich | Travel time | Brenner Pass | Brenner base tunnel |
| | Monetary cost | Brenner Pass | Brenner base tunnel |
| | Generalized cost | Brenner Pass | Brenner base tunnel |
| Berlin/ Munich—Leghorn | Travel time | Brenner Pass | Brenner base tunnel |
| | Monetary cost | Brenner Pass | Brenner base tunnel |
| | Generalized cost | Brenner Pass | Brenner base tunnel |
| Venice—Munich | Travel time | Tarvisio, Tauern | Brenner base tunnel |
| | Monetary cost | Tarvisio, Tauern | Brenner base tunnel |
| | Generalized cost | Tarvisio, Tauern | Brenner base tunnel |
| Munich—Venice | Travel time | Tauern, Tarvisio | Brenner base tunnel |
| | Monetary cost | Tauern, Tarvisio | Brenner base tunnel |
| | Generalized cost | Tauern, Tarvisio | Brenner base tunnel |
| Leghorn—Prague | Travel time | Tarvisio, Tauern | Brenner base tunnel |
| | Monetary cost | Tarvisio, Tauern | Brenner base tunnel |
| | Generalized cost | Tarvisio, Tauern | Brenner base tunnel |
| Prague—Leghorn | Travel time | Tauern, Tarvisio | Brenner base tunnel |
| | Monetary cost | Tauern, Tarvisio | Brenner base tunnel |
| | Generalized cost | Tauern, Tarvisio | Brenner base tunnel |
| Milan—Vienna | Travel time | Tarvisio, Klagenfurt—Bruck old line, Semmering Pass | Tarvisio, Koralmbahn, Semmering base tunnel |
| | Monetary cost | Tarvisio, Klagenfurt—Bruck old line, Semmering Pass | Tarvisio, Koralmbahn, Semmering base tunnel |
| | Generalized cost | Tarvisio, Klagenfurt—Bruck old line, Semmering Pass | Tarvisio, Koralmbahn, Semmering base tunnel |
| Vienna—Milan | Travel time | Semmering Pass, Bruck—Klagenfurt old line, Tarvisio | Semmering base tunnel, Koralmbahn, Tarvisio |
| | Monetary cost | Semmering Pass, Bruck—Klagenfurt old line, Tarvisio | Semmering base tunnel, Koralmbahn, Tarvisio |
| | Generalized cost | Semmering Pass, Bruck—Klagenfurt old line, Tarvisio | Semmering base tunnel, Koralmbahn, Tarvisio |

**Table A1.** *Cont.*

| O–D Pair | | Current Scenario | Project Scenario |
|---|---|---|---|
| Milan—Budapest | Travel time | Tarvisio, Klagenfurt—Bruck old line, Semmering Pass | Tarvisio, Koralmbahn, Semmering base tunnel |
| | Monetary cost | Villa Opicina, Divaca, Ljubljana, Ormoz | Villa Opicina, Divaca, Ljubljana, Ormoz |
| | Generalized cost | Villa Opicina, Divaca, Ljubljana, Ormoz | Villa Opicina, Divaca, Ljubljana, Ormoz |
| Budapest—Milan | Travel time | Semmering Pass, Bruck—Klagenfurt old line, Tarvisio | Semmering base tunnel, Koralmbahn, Tarvisio |
| | Monetary cost | Ormoz, Ljubljana, Divaca, Villa Opicina | Ormoz, Ljubljana, Divaca, Villa Opicina |
| | Generalized cost | Ormoz, Ljubljana, Divaca, Villa Opicina | Ormoz, Ljubljana, Divaca, Villa Opicina |
| Trieste—Vienna | Travel time | Tarvisio, Klagenfurt-Bruck old line, Semmering Pass | Tarvisio, Koralmbahn, Semmering base tunnel |
| | Monetary cost | Villa Opicina, Divaca, Ljubljana, Maribor, Semmering Pass | New Trieste—Divaca line, Ljubljana, Maribor, Semmering base tunnel |
| | Generalized cost | Tarvisio, Klagenfurt-Bruck old line, Semmering Pass | Tarvisio, Koralmbahn, Semmering base tunnel |
| Vienna—Trieste | Travel time | Semmering Pass, Bruck-Klagenfurt old line, Tarvisio | Semmering base tunnel, Koralmbahn, Tarvisio |
| | Monetary cost | Semmering Pass, Maribor, Ljubljana, Divaca, Villa Opicina | Semmering base tunnel, Maribor, Ljubljana, new Divaca—Trieste line |
| | Generalized cost | Semmering Pass, Bruck-Klagenfurt old line, Tarvisio | Semmering base tunnel, Koralmbahn, Tarvisio |
| Vienna—Koper | Travel time | Semmering Pass, Bruck-Klagenfurt old line, Tarvisio, Villa Opicina, Divaca-Koper old line | Semmering base tunnel, Koralmbahn, Tarvisio, Villa Opicina, Divaca-Koper new line |
| | Monetary cost | Semmering Pass, Maribor, Ljubljana, Divaca-Koper old line | Semmering base tunnel, Maribor, Ljubljana, Divaca-Koper new line |
| | Generalized cost | Semmering Pass, Maribor, Ljubljana, Divaca-Koper old line | Semmering base tunnel, Maribor, Ljubljana, Divaca-Koper new line |
| Koper—Vienna | Travel time | Koper-Divaca old line, Villa Opicina, Tarvisio, Klagenfurt-Bruck old line, Semmering Pass | Koper-Divaca new line, Villa Opicina, Tarvisio, Koralmbahn, Semmering base tunnel |
| | Monetary cost | Koper-Divaca old line, Ljubljana, Maribor, Semmering Pass | Koper-Divaca new line, Ljubljana, Maribor, Semmering base tunnel |
| | Generalized cost | Koper-Divaca old line, Ljubljana, Maribor, Semmering Pass | Koper- Divaca new line, Ljubljana, Maribor, Semmering base tunnel |

**Table A1.** *Cont.*

| O–D Pair | | Current Scenario | Project Scenario |
|---|---|---|---|
| Hamburg—Zagreb | Travel time | Tauern, Karavanke, Ljubljana | Semmering base tunnel, Graz-Maribor line |
| | Monetary cost | Tauern, Karavanke, Ljubljana | Semmering base tunnel, Graz-Maribor line |
| | Generalized cost | Tauern, Karavanke, Ljubljana | Semmering base tunnel, Graz-Maribor line |
| Zagreb—Hamburg | Travel time | Ljubljana, Karavanke, Tauern | Maribor-Graz line, Semmering base tunnel |
| | Monetary cost | Ljubljana, Karavanke, Tauern | Maribor-Graz line, Semmering base tunnel |
| | Generalized cost | Ljubljana, Karavanke, Tauern | Maribor-Graz line, Semmering base tunnel |
| Vienna—Lyon | Travel time | Not through the Alps but via Munich, Bregenz, Zurich, Bern, Lausanne | Semmering base tunnel, Koralmbahn, Tarvisio, new Frejus base tunnel |
| | Monetary cost | Semmering Pass, Bruck-Klagenfurt old line, Tarvisio, Frejus old tunnel | Semmering base tunnel, Koralmbahn, Tarvisio, new Frejus base tunnel |
| | Generalized cost | Not through the Alps but via Munich, Bregenz, Zurich, Bern, Lausanne | Semmering base tunnel, Koralmbahn, Tarvisio, new Frejus base tunnel |
| Lyon—Vienna | Travel time | Not through the Alps but via Lausanne, Bern, Zurich, Bregenz, Munich | New Frejus base tunnel, Tarvisio, Koralmbahn, Semmering base tunnel |
| | Monetary cost | Frejus old line, Tarvisio, Klagenfurt—Bruck old line, Semmering Pass | New Frejus base tunnel, Tarvisio, Koralmbahn, Semmering base tunnel |
| | Generalized cost | Not through the Alps but via Lausanne, Bern, Zurich, Bregenz, Munich | New Frejus base tunnel, Tarvisio, Koralmbahn, Semmering base tunnel |
| Munich—Avignon rail junction | Travel time | Not through the Alps but via Bregenz, Zurich, Bern, Lausanne | Brenner base tunnel, new Frejus base tunnel |
| | Monetary cost | Not through the Alps but via Bregenz, Zurich, Bern, Lausanne | Brenner base tunnel, new Frejus base tunnel |
| | Generalized cost | Not through the Alps but via Bregenz, Zurich, Bern, Lausanne | Brenner base tunnel, new Frejus base tunnel |
| Avignon rail junction—Munich | Travel time | Not through the Alps but via Lausanne, Bern, Zurich, Bregenz | New Frejus base tunnel, Brenner base tunnel |
| | Monetary cost | Not through the Alps but via Lausanne, Bern, Zurich, Bregenz | New Frejus base tunnel, Brenner base tunnel |
| | Generalized cost | Not through the Alps but via Lausanne, Bern, Zurich, Bregenz | New Frejus base tunnel, Brenner base tunnel |

**Table A1.** *Cont.*

| O–D Pair | | Current Scenario | Project Scenario |
|---|---|---|---|
| Munich—Marseilles | Travel time | Brenner Pass, Giovi Pass, Ventimiglia | Brenner base tunnel, Third Pass of Giovi, Ventimiglia |
| | Monetary cost | Brenner Pass, Giovi Pass, Ventimiglia | Brenner base tunnel, Third Pass of Giovi, Ventimiglia |
| | Generalized cost | Brenner Pass, Giovi Pass, Ventimiglia | Brenner base tunnel, Third Pass of Giovi, Ventimiglia |
| Marseilles—Munich | Travel time | Ventimiglia, Giovi Pass, Brenner Pass | Ventimiglia, Third Pass of Giovi, Brenner base tunnel |
| | Monetary cost | Ventimiglia, Giovi Pass, Brenner Pass | Ventimiglia, Third Pass of Giovi, Brenner base tunnel |
| | Generalized cost | Ventimiglia, Giovi Pass, Brenner Pass | Ventimiglia, Third Pass of Giovi, Brenner base tunnel |

## Appendix B

**Table A2.** Travel times [h], monetary costs [€/train] and generalized costs [€/train], in the 'current scenario' and in the 'project scenario', ordered by the considered O–D pairs.

| O–D Pair | Travel Time (h) | | Monetary Cost (€/train) | | Generalized Cost (€/train) | |
|---|---|---|---|---|---|---|
| | Current Scenario | Project Scenario | Current Scenario | Project Scenario | Current Scenario | Project Scenario |
| Turin—Paris | 11.14 | 10.34 | 15,553.7 | 14,788.0 | 23,049.5 | 21,744.5 |
| Paris—Turin | 11.07 | 10.33 | 15,492.3 | 14,782.2 | 22,994.5 | 21,739.5 |
| Genoa—Paris | 13.40 | 12.26 | 17,941.1 | 16,866.4 | 26,967.5 | 25,113.5 |
| Paris—Genoa | 13.33 | 12.25 | 17,898.4 | 16,833.5 | 26,928.5 | 25,083.5 |
| Leghorn—Paris | 16.30 | 15.16 | 20,398.3 | 19,323.6 | 31,431.5 | 29,577.5 |
| Paris—Leghorn | 16.27 | 15.11 | 20,355.7 | 19,290.7 | 31,392.5 | 29,547.5 |
| Milan—Avignon rail junction | 8.97 | 8.17 | 13,186.1 | 12,420.4 | 19,236.5 | 17,931.5 |
| Avignon rail junction—Milan | 8.93 | 8.15 | 13,124.7 | 12,414.6 | 19,181.5 | 17,926.5 |
| Turin—Rotterdam | 16.87 | 16.28 | 21,347.1 | 20,581.4 | 33,261.5 | 31,956.5 |
| Rotterdam—Turin | 16.83 | 16.25 | 21,285.7 | 20,575.7 | 33,206.5 | 31,951.5 |
| Turin—Antwerp | 15.83 | 15.24 | 19,694.8 | 18,929.1 | 30,577.5 | 29,272.5 |
| Antwerp—Turin | 15.79 | 15.24 | 19,633.4 | 18,923.4 | 30,522.5 | 29,267.5 |
| Novara—Paris | 11.99 | 11.54 | 16,938.9 | 16,173.2 | 25,014.5 | 23,929.5 |
| Paris—Novara | 11.93 | 11.52 | 16,877.5 | 16,167.5 | 24,998.5 | 23,924.5 |
| Novara—Antwerp | 14.67 | 14.67 | 19,281.3 | 19,281.3 | 29,495.5 | 29,495.5 |
| Antwerp—Novara | 14.64 | 14.64 | 18,994.7 | 18,994.7 | 28,918.5 | 28,918.5 |
| Novara—Rotterdam | 15.71 | 15.71 | 20,933.4 | 20,933.4 | 32,179.5 | 32,179.5 |
| Rotterdam—Novara | 15.68 | 15.68 | 20,647.2 | 20,647.2 | 31,602.5 | 31,602.5 |
| Turin—Hamburg | 18.55 | 18.55 | 27,055.3 | 27,055.3 | 39,137.5 | 39,137.5 |
| Hamburg—Turin | 18.52 | 18.52 | 27,048.9 | 27,048.9 | 39,121.5 | 39,121.5 |
| Milan—Paris | 12.18 | 11.98 | 17,692.9 | 17,080.2 | 25,773.5 | 25,067.5 |
| Paris—Milan | 12.15 | 11.97 | 17,613.2 | 16,986.1 | 25,691.5 | 24,964.5 |
| Milan—Antwerp | 14.56 | 14.36 | 19,462.7 | 18,849.9 | 29,273.5 | 28,567.5 |
| Antwerp—Milan | 14.50 | 14.33 | 19,382.9 | 18,755.9 | 29,191.5 | 28,464.5 |
| Milan—Rotterdam | 15.60 | 15.40 | 21,115.0 | 20,502.2 | 31,957.5 | 31,251.5 |
| Rotterdam—Milan | 15.56 | 15.37 | 21,035.2 | 20,408.2 | 31,875.5 | 31,148.5 |
| Milan—Duisburg | 12.81 | 12.61 | 20,290.5 | 19,677.7 | 28,590.5 | 27,884.5 |
| Duisburg—Milan | 12.75 | 12.55 | 20,210.7 | 19,583.6 | 28,508.5 | 27,781.5 |
| Milan—Hamburg | 17.28 | 17.08 | 26,254.0 | 25,641.2 | 37,430.5 | 36,724.5 |
| Hamburg—Milan | 17.24 | 17.05 | 26,174.2 | 25,547.1 | 37,348.5 | 36,621.5 |
| Milan—Berlin | 16.59 | 16.22 | 25,060.2 | 24,130.1 | 36,248.5 | 34,679.5 |
| Berlin—Milan | 16.53 | 16.17 | 24,839.4 | 23,987.2 | 36,051.5 | 34,493.5 |
| Genoa—Zurich | 6.05 | 5.57 | 10,476.1 | 10,143.7 | 14,542.5 | 13,860.5 |
| Zurich—Genoa | 6.06 | 5.54 | 10,488.4 | 10,110.1 | 14,542.5 | 13,819.5 |
| Turin—Munich | 10.52 | 9.84 | 16,014.0 | 15,084.0 | 22,976.5 | 21,611.5 |

**Table A2.** *Cont*.

| O–D Pair | Travel Time (h) | | Monetary Cost (€/train) | | Generalized Cost (€/train) | |
|---|---|---|---|---|---|---|
| | Current Scenario | Project Scenario | Current Scenario | Project Scenario | Current Scenario | Project Scenario |
| Munich—Turin | 10.48 | 9.82 | 15,793.2 | 14,941.1 | 22,960.5 | 21,425.5 |
| Milan—Munich | 8.30 | 7.36 | 13,549.5 | 12,619.5 | 19,032.5 | 17,463.5 |
| Munich—Milan | 8.04 | 7.31 | 13,328.7 | 12,476.5 | 18,835.5 | 17,277.5 |
| Livorno—Berlin | 19.23 | 18.29 | 27,161.9 | 26,231.8 | 39,759.5 | 38,190.5 |
| Berlin—Livorno | 19.18 | 18.25 | 26,941.0 | 26,088.9 | 39,562.5 | 38,004.5 |
| Livorno—Munich | 10.37 | 9.43 | 15,651.2 | 14,721.1 | 22,543.5 | 20,974.5 |
| Munich—Livorno | 10.26 | 9.37 | 15,430.3 | 14,578.2 | 22,346.5 | 20,788.5 |
| Venice—Munich | 7.53 | 7.08 | 13,088.2 | 12,158.2 | 18,257.5 | 16,829.5 |
| Munich—Venice | 7.47 | 7.01 | 12,867.4 | 12,015.3 | 18,201.5 | 16,643.5 |
| Leghorn—Prague | 16.62 | 15.68 | 23,573.3 | 22,643.2 | 34,514.5 | 32,945.5 |
| Prague—Leghorn | 16.57 | 15.61 | 23,352.4 | 22,500.3 | 34,317.5 | 32,759.5 |
| Milan—Vienna | 11.32 | 10.18 | 16,727.6 | 16,226.5 | 24,349.5 | 22,932.5 |
| Vienna—Milan | 11.48 | 10.21 | 16,806.5 | 16,270.7 | 24,419.5 | 22,972.5 |
| Milan—Budapest | 14.46 | 13.22 | 18,074.1 | 18,074.1 | 28,797.5 | 27,380.5 |
| Budapest—Milan | 14.51 | 13.25 | 18,125.3 | 18,125.3 | 28,867.5 | 27,420.5 |
| Trieste—Vienna | 8.14 | 6.90 | 12,107.5 | 11,450.2 | 18,141.5 | 16,724.5 |
| Vienna- Trieste | 8.16 | 6.92 | 12,134.2 | 11,400.3 | 18,211.5 | 16,764.5 |
| Vienna—Koper | 9.61 | 8.41 | 12,287.9 | 11,357.2 | 19,620.5 | 17,340.5 |
| Koper—Vienna | 9.69 | 8.49 | 12,340.6 | 11,401.2 | 19,667.5 | 17,379.5 |
| Hamburg—Zagreb | 21.06 | 20.18 | 28,858.6 | 28,597.6 | 42,684.5 | 42,461.5 |
| Zagreb—Hamburg | 21.13 | 20.22 | 29,040.8 | 28,836.2 | 42,848.5 | 42,607.5 |
| Vienna—Lyon | 18.08 | 16.89 | 24,347.8 | 23,046.3 | 36,996.5 | 34,244.5 |
| Lyon—Vienna | 17.99 | 16.86 | 24,207.5 | 22,996.3 | 36,871.5 | 34,199.5 |
| Munich—Avignon rail junction | 15.93 | 15.45 | 21,636.4 | 20,834.4 | 32,675.5 | 31,146.5 |
| Avignon rail junction—Munich | 15.99 | 15.52 | 21,652.9 | 20,971.6 | 32,675.5 | 31,327.5 |
| Munich—Marseilles | 15.96 | 14.78 | 20,882.5 | 19,764.5 | 31,757.5 | 29,832.5 |
| Marseilles—Munich | 16.25 | 14.95 | 21,084.6 | 19,934.6 | 31,938.5 | 30,043.5 |

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
