# Peer review of "Railway Lines across the Alps: Analysis of Their Usage through a New Railway Link Cost Function"

_applsci, doi:10.3390/app10093120_

Round 1

Reviewer 1 Report

1. An abstract should be written shorter and more concentrated. It is especially important to reflect scientific novelty.

2. The dimensions in formula (1) should be checked, since it is inconsistent between the individual terms. For example in nw ∙ (Aw + Mw + Iw) + R : Aw - cost (EUR); Mw, Iw similarly; nw - dimensionless quantity; R - railway track cost (EUR/km).

3. Table 1 is excessively long (5 pages!). It is enough to give the most important routes as an example.

4. Table 2 may also be reduced.

Author Response

Please find our responses in the attached file.

Reviewer 2 Report

The Authors wrote about evaluation by a new cost function, but:
- there is no structured literature review that would show what scientific gap exists in that area – must be done,
- advantages and disadvantages of existing methods must be shown.

The proposed cost function (formula 1) must be described more in details. It is not sufficient to explain the lettering in point form. Moreover, the new component, introduced during the performed research must be highlighted.

Acceleration and deceleration is important in cost generating. This issue depends on the time table, which may be worse for cargo trains in case of mixed, heterogeneous traffic. I can’t see taking into account of that issues. I suggest to introduce them, or to highlight them.

In chapter 4., the used methods were not described. How were found the optimal paths? Description should be supported by appropriate subchapters in the literature review.

The chapter titles should be renamed.

Table 1. is too long. It should be shown in form of a few rows as an example, and the whole table should move to the end of the paper as an appendix. The same in terms of table 2.

The conclusions should be focussed on discussion of advantages and disadvantages of the proposed cost function and method, as well as further research in terms of them.

Author Response

Please find our responses in the attached file

Reviewer 3 Report

Very interesting report. I recommend to Authors the extension of the literature review on the subject discussed. I would suggest analyzing the work of other authors than the authors of the article.

Author Response

Very interesting report. I recommend to Authors the extension of the literature review on the subject discussed. I would suggest analyzing the work of other authors than the authors of the article.

Thank you. We improved the literature review: sub-section 2.1.1.

Thank you very much for your revision.

Reviewer 4 Report

The presented study is at the required level. I recommend publishing it.

I have one note that is in the line 585 -857. Maybe exists relevant studies about costs problematics in ralway transport. For examle you can check article:

Dolinayova, A. Loch, M., Kanis. J. (2015). Modelling the influence of wagon technical parameters on variable costs in rail freight transport. RESEARCH IN TRANSPORTATION ECONOMICS. Vol. 54 p. 33-40

or similar relevants journals.

Author Response

The presented study is at the required level. I recommend publishing it. I have one note that is in the line 585 -857. Maybe exists relevant studies about costs problematics in railway transport.

For example you can check article:

Dolinayova, A. Loch, M., Kanis. J. (2015). Modelling the influence of wagon technical parameters on variable costs in rail freight transport. RESEARCH IN TRANSPORTATION ECONOMICS. Vol. 54 p. 33-40

or similar relevants journals.

Thank you. We improved the literature review: sub-section 2.1.1. In particular we added to the literature review the suggested article.

Thank you very much for your revision.